# Length Optimization in Conformal Prediction

**Shayan Kiyani, George Pappas, Hamed Hassani**
Department of Electrical and Systems Engineering
University of Pennsylvania
{shayank, pappasg, hassani}@seas.upenn.edu

## Abstract

Conditional validity and length efficiency are two crucial aspects of conformal prediction (CP). Conditional validity ensures accurate uncertainty quantification for data subpopulations, while proper length efficiency ensures that the prediction sets remain informative. Despite significant efforts to address each of these issues individually, a principled framework that reconciles these two objectives has been missing in the CP literature. In this paper, we develop Conformal Prediction with Length-Optimization (CPL) - a novel and practical framework that constructs prediction sets with (near-) optimal length while ensuring conditional validity under various classes of covariate shifts, including the key cases of marginal and group-conditional coverage. In the infinite sample regime, we provide strong duality results which indicate that CPL achieves conditional validity and length optimality. In the finite sample regime, we show that CPL constructs conditionally valid prediction sets. Our extensive empirical evaluations demonstrate the superior prediction set size performance of CPL compared to state-of-the-art methods across diverse real-world and synthetic datasets in classification, regression, and large language model-based multiple choice question answering. An Implementation of our algorithm can be accessed at the following link: https://github.com/shayankiyani98/CP.

## 1 Introduction

Consider a distribution $\mathcal{D}$ over the domain $\mathcal{X} \times \mathcal{Y}$, where $\mathcal{X}$ is the covariate space and $\mathcal{Y}$ is the label space. Using a set of calibration samples $(X_1, Y_1), \ldots, (X_n, Y_n)$ drawn i.i.d. from $\mathcal{D}$, the objective of conformal prediction (CP) is to create a prediction set $C(x)$, for each input $x$, that is likely to include the true label $y$. This is formalized through specific coverage guarantees on the prediction sets. For example, the simplest, and yet the most commonly-used guarantee is the *marginal coverage*: The prediction sets $C(x) \subseteq \mathcal{Y}$ achieve marginal coverage if, for a test sample $(X_{n+1}, Y_{n+1})$, we have $\Pr(Y_{n+1} \in C(X_{n+1})) = 1 - \alpha$. Here, $\alpha$ is the given miscoverage rate, and the probability is over the randomness in the calibration and test points.

Conformal Prediction faces two major challenges in practice: conditional validity and length inefficiency. Marginal coverage is often a weak guarantee; in many practical scenarios we need coverage guarantees that hold across different sub-populations of the data. E.g., in healthcare applications, obtaining valid prediction sets for different patient demographics is crucial; we often need to construct accurate prediction sets for certain age groups or medical conditions. This is known as conditional validity - which ideally seeks a property called *full conditional coverage*: For every $x \in \mathcal{X}$ we require

$$\Pr\left(Y_{n+1} \in C\left(X_{n+1}\right) \mid X_{n+1} = x\right) = 1 - \alpha. \tag{1}$$

Alas, achieving full conditional coverage with finite calibration data is known to be impossible [1–3]. Consequently, in recent years several algorithmic frameworks have been developed that guarantee relaxations of (1)[4–19]. However, the prediction sets constructed by these methods are often observed to be (unnecessarily) large in size [2, 20, 21]; i.e., it is possible to find other conditionally-valid prediction sets with smaller length. Even in the case of marginal coverage, the

38th Conference on Neural Information Processing Systems (NeurIPS 2024).

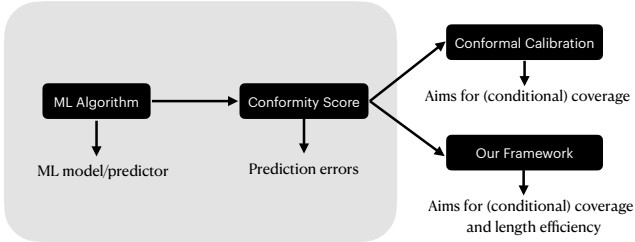

Figure 1: The CP pipeline.

prediction sets constructed by algorithms such as split-conformal can be far from optimal in size. This brings us to the second major challenge of CP, namely length efficiency.

Length efficiency is about constructing prediction sets whose size (or length) is as small as possible while maintaining conditional validity. Here, "length" refers to the average length of the prediction sets $C(X)$ over the distribution of $X$. Length efficiency is crucial for the prediction sets to be informative [22–25]. In fact, the performance of CP methods are closely tied to the length efficiency of the prediction sets in practical applications in decision making[26–29], robotics[20, 21], communication systems [30, 31], and large language models [32–34]. This raises the question of how length and coverage fundamentally interact. Along this line, we ask: *How can we construct prediction sets that are (near-) optimal in length while maintaining valid (conditional) coverage?*

In this paper, we answer this question by proposing a unified framework for length optimization under coverage constraints. Before providing the details, we find it necessary to explain (i) where in the CP pipeline our framework is operating, and (ii) the crucial role of the covariate $X$.

First, we note that the pipeline of CP consists of three stages (see Figure 1) which are often treated separately [8]: Training a model using training data, designing a conformity score, and constructing the prediction sets. Our framework operates at the third stage, i.e., we aim at designing the prediction sets assuming a given predictive model as well as a conformity score. In recent years, various powerful frameworks have been developed for designing better conformity scores [22, 24, 35–41] and obtaining conditional guarantees using a given score [4–19, 42, 43]. The missing piece in this picture is length optimization which is the subject of this paper.

Second, we emphasize that length optimization can be highly dependent on the structural properties of the covariate $X$. It has been known in the CP community that the structure of $X$ can play a role in terms of length efficiency and coverage validity (see e.g. [22, 35]). However, to our knowledge, a principled study of length optimization using the covariate $X$ is missing. To showcase the principles and challenges of using the covariate $X$ in the design of prediction sets, we provide a toy example.

**Example.** Let $\mathcal{X} = [-1, +1]$. Assume $X$ is uniformly distributed over $\mathcal{X}$, and $Y$ is distributed as:

$$\text{-if } x < 0, \text{ then } Y \sim x + \mathcal{N}(0, 2), \qquad \text{and} \qquad \text{-if } x \geq 0, \text{ then } Y \sim x + \mathcal{N}(0, 1),$$

see Figure 2-(a). For simplicity, we assume in this example that we have infinitely many data points available from this distribution. As the model, we consider $f(x) = \mathbb{E}[Y \mid X = x] = x$. As a result, considering the conformity score $S(x, y) = |y - f(x)|$, the distribution of $S$ follows the folded Gaussian distribution: i.e. for $x < 0$ we have $\mathcal{D}_{S|x} = |N(0, 2)|$, and for $x \geq 0$ we have $\mathcal{D}_{S|x} = |N(0, 1)|$ – see Figure 2-(b). We aim for marginal coverage of 0.9, and consider prediction sets $C(x)$ in the following form:

-if $x < 0 : C(x) = \{y \in \mathbb{R} \text{ s.t. } S(x, y) \leq q_-\}$, and - if $x \geq 0 : C(x) = \{y \in \mathbb{R} \text{ s.t. } S(x, y) \leq q_+\}$.

For every value of $q_+$ in the range $[1.4, +\infty]$ there exists a unique $q_-$ that ensures 0.9-marginal coverage. This provides a continuous *family of prediction sets*, parameterized by $q_+$, all of which are marginally valid, but have different average lengths. Length optimization over this family amounts to minimizing the average length (which is equal to $q_- + q_+$) over the choice of $q_+$. Figure 2-(c) plots the average length versus $q_+$. Four lessons from this example are as follows: (i) First, as we see from Figure 2-(c), the optimal-length solution is different from the sets constructed by the Split-Conformal method (for which $q_-$ and $q_+$ are both equal to the 0.9-quantile of the marginal distribution of $S$). Hence, the structure of the optimal prediction sets *depends on the structure of the covariates* (in this case the sign of the covariate). It can be shown that the optimal-solution found is the unique globally optimal-length solution among all the possible marginally-valid prediction

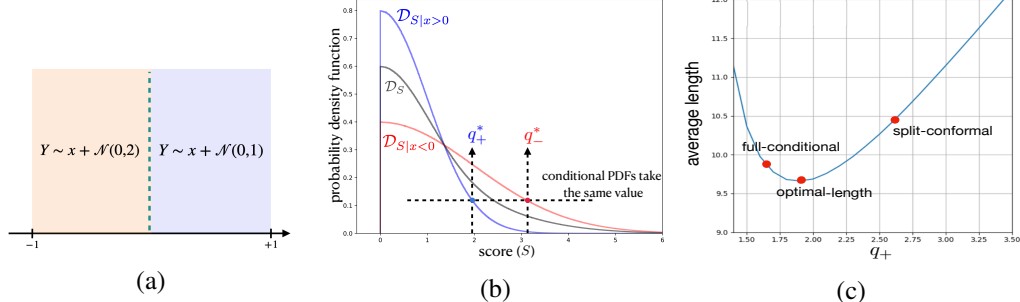

(a)  (b)  (c)

Figure 2: (a) Distribution of the labels conditioned on the covariate $x$ (b) The conditional PDFs. (c) Avg length vs $q_+$. The red dots correspond to three different marginally-valid prediction sets.

sets. Hence, the feature $h(x) = \text{sign}(x)$ is crucial in determining the optimal-length sets. (ii) Second, for $(q_-^*, q_+^*)$ corresponding to the optimal-length sets, the conditional PDFs take the same value; i.e. $p(S = q_+^* | x \geq 0) = p(S = q_-^* | x \leq 0)$ – see Figure 2-(b). This is not a coincidence, and as we will show, it is the main deriving principle to find the optimal sets – see Example 3.2 below.

**Contributions.** We develop a principled and practical framework to design length-optimized prediction sets that are conditionally valid. We formalize conditional validity using a class of covariate shifts (as in [8]). As for the class, we consider a linear span of finitely-many basis functions of the covariates. This is a broad class; E.g., by adjusting the basis functions we can recover marginal coverage or group-conditional coverage as special cases.

We develop a foundational minimax notion where the min part ensures conditional validity and the max part optimizes length. In the infinite-sample setting, the solution of the minimax problem is proved to be the optimal-length prediction sets, through a strong duality result. To design our practical algorithm for the finite-sample setting (termed as CPL), our key insight is to relax minimax problem using a given conformity score and a hypothesis class that aims to learn from data features that are important for length optimization and uncertainty quantification. In the finite sample setting, we guarantee that our algorithm remains conditionally valid up to a finite-sample error.

We extensively evaluate the performance of CPL on a variety of real world and synthetic datasets in classification, regression, and text-based settings against numerous state-of-the-art algorithms ranging from Split-Conformal[22, 44], Jacknife[45], Local-CP[12], and CQR[35], to BatchGCP[4] and Conditional-Calibration[8]. A detailed discussion of our experiments in section 5 showcase the overall superior performance of CPL in achieving significantly better length efficiency in different scenarios where we demand marginal validity, group conditional validity, or the more complex case of conditional validity with respect to a class of covariate shifts.

## 2 Problem Formulation

### 2.1 Preliminaries on conditional validity and length of prediction sets

Recall that $(X, Y)$ is generated from a distribution $\mathcal{D}$ supported on $\mathcal{X} \times \mathcal{Y}$ (distributions $\mathcal{D}_X$ and $\mathcal{D}_{Y|X}$ are then defined naturally). The prediction sets are of the form $C(x) : \mathcal{X} \to 2^{\mathcal{Y}}$ for $x \in \mathcal{X}$.

**Conditional Coverage Validity.** Ultimately, in conformal prediction, the goal is to design prediction sets that satisfy the *full conditional coverage* introduced in (1). Despite the importance, full conditional coverage is impossible to achieve in the finite-sample setting [1–3]. Therefore, several weaker notions of coverage such as marginal [22, 44], group conditional [4–6], and coverage with respect to a class of covariate shifts [8–10] have been considered in the literature. Here, we focus on the notion of conditional validity with respect to a class of covariate shifts, which unifies all these notions, and contains each of them as a special case by adjusting the class. Fix any non-negative function $f$ and let $\mathcal{D}_f$ denote the setting in which $\{(X_i, Y_i)\}_{i=1}^n$ is sampled i.i.d. from $\mathcal{D}$, while $(X_{n+1}, Y_{n+1})$ is sampled independently from the distribution in which $\mathcal{D}_X$ is *shifted* by $f$, i.e.,

$$X_{n+1} \sim \frac{f(x)}{\mathbb{E}_{\mathcal{D}}[f(X)]} \cdot d\mathcal{D}_X(x), \quad Y_{n+1} \mid X_{n+1} \sim \mathcal{D}_{Y|X}.$$

We define the exact coverage validity under non-negative *covariate shift* $f$ as, $\Pr_{X_{n+1} \sim \mathcal{D}_f} (Y_{n+1} \in C(X_{n+1})) = 1 - \alpha$, which can be equivalently written as,

$$\mathbb{E}_{X_{n+1} \sim \mathcal{D}} \left[ f(X_{n+1}) \Big\{ \mathbf{1}[Y_{n+1} \in C(X_{n+1})] - (1 - \alpha) \Big\} \right] = 0. \tag{2}$$

This last equality enables us to define the notion of generalized covariate shift. For prediction sets $C$, we say that coverage with respect to a function $f$ (which can take negative values) is guaranteed if (2) holds. We often drop the term "generalized" when we refer to generalized covariate shifts.

Let $\mathcal{F}$ be a class of functions from $\mathcal{X}$ to $\mathbb{R}$. We say the prediction sets $C$ satisfy valid conditional coverage with respect to the class of covariate shifts $\mathcal{F}$, if (2) holds for every $f \in \mathcal{F}$.

It is easy to see that if $\mathcal{F}$ is the class of all (measurable) functions on $\mathcal{X}$, then conditional validity w.r.t. $\mathcal{F}$ is equivalent to full conditional converge (1). Accordingly, using less complex choices for $\mathcal{F}$ would result in relaxed notions of conditional coverage. We now review three important instances of $\mathcal{F}$:

1) **Marginal Coverage:** Setting $\mathcal{F} = \{x \mapsto c \mid c \in \mathbb{R}\}$, i.e. constant functions, recovers the marginal validity, i.e., $\Pr_{X_{n+1} \sim \mathcal{D}}(Y_{n+1} \in C(X_{n+1})) = 1 - \alpha$.

2) **Group-Conditional Coverage:** Let $G_1, G_2, \cdots, G_m \subseteq \mathcal{X}$ be a collection of sub-groups in the covariate space. Define $f_i(x) = \mathbf{1}[x \in G_i]$ for every $i \in [1, m]$. By using the class of covariate shifts $\mathcal{F} = \{\sum_{i=1}^m \beta_i f_i(x) \mid \beta_i \in \mathbb{R} \text{ for every } i \in [1, \cdots, m]\}_{i=1}^m$ we can obtain the group-conditional coverage validity; i.e. $\Pr_{X_{n+1} \sim \mathcal{D}}(Y_{n+1} \in C(X_{n+1})|X \in G_i) = 1 - \alpha$, for every $i \in [1, m]$.

3) **Finite-dimensional Affine Class of Covariate Shifts:** Let $\Phi : \mathcal{X} \to \mathbb{R}^d$ be a *predefined* function (a.k.a. the finite-dimensional basis). The class $\mathcal{F}$ is defined as $\mathcal{F} = \{\langle \boldsymbol{\beta}, \Phi(x) \rangle | \boldsymbol{\beta} \in \mathbb{R}^d\}$. Here, $\mathcal{F}$ is an affine class of covariate shifts as for any $f, f' \in \mathcal{F}$ and $\epsilon \in \mathbb{R}$ we have $f + \epsilon f' \in \mathcal{F}$. We use the term "bounded" class of covariate shifts when we assume, $\max_{1 \le i \le d} \sup_{x \in \mathcal{X}} \phi_i(x) < \infty$, where $\Phi(x) = [\phi_1(x), \cdots, \phi_d(x)]$.

The finite-dimensional affine class is fairly general and covers a broad range of scenarios in theory and practice (see Section 5.3, Remark D.1 in appendix D, as well as [8]). As special cases, it includes both of the marginal and group-conditional settings. To see this, one can pick $\Phi(x) = 1$ for the marginal case and $\Phi = [f_1(x), f_2(x), \cdots, f_m(x)]^T$ for the group-conditional scenario. Consequently, from now on we will focus on the scenario where $\mathcal{F}$ is a $d$-dimensional affine class of covariate shifts.

**Length.** We use the term $\text{len}(C(x))$ to denote the length of a prediction set at point $x \in \mathcal{X}$. Length can have different interpretations in different contexts, yet it always depicts the same intuitive meaning of size. In the case of regression when $\mathcal{Y} = \mathbb{R}$ we have, $\text{len}(C(x)) = \int_{\mathcal{Y}} \mathbf{1}[y \in C(x)]dy$. Here, $dy$ can be interpreted as Lebesgue integral (i.e. the usual way to measure length on $\mathbb{R}$). Similarly, in the classification setting, when $\mathcal{Y} = \{y_1, y_2, \cdots, y_L\}$, we let $\text{len}(C(x)) = \sum_{i=1}^L \mathbf{1}[y \in C(x)]$.

### 2.2 Problem Statement

In the previous section, we defined conditional validity and length as two main notions for prediction sets. The canonical problem that we study in this paper is:

---

**Primary Problem:**

$$\underset{C(x)}{\text{Minimize}} \quad \mathbb{E}_X\left[\text{len}(C(X))\right]$$

$$\text{subject to} \quad \mathbb{E}_{X,Y}\left[f(X)\left\{\mathbf{1}[Y \in C(X)] - (1 - \alpha)\right\}\right] = 0, \quad \forall f \in \mathcal{F}$$

---

I.e., we want to construct predictions sets $C(x)$, $x \in \mathcal{X}$ with optimal (i.e. minimal) average length while being conditionally valid with respect to a (finite-dimensional) class of covariate shifts $\mathcal{F}$.

In the special case of marginal validity the constraint becomes $\mathbb{E}\left[c\left\{\mathbf{1}[Y \in C(X)] - (1 - \alpha)\right\}\right] = 0$ for every $c \in \mathbb{R}$, or equivalently $\Pr(Y \in C(X)) = 1 - \alpha$. Similarly, in the special case of group-conditional the constraint boils down to $\Pr(Y \in C(X)|X \in G_i) = 1 - \alpha$ for every $i \in [1, \cdots, m]$.

## 3 Minimax Formulations

In this section we analyze the Primary Problem 2.2 in the infinite-data regime. Our goal is to derive the principles of an algorithmic framework for Primary Problem 2.2. We will do so by deriving an equivalent minimax formulation whose solutions have a rich interpretation in terms of level sets of the data distribution. Having the practical finite-sample setting in mind, we then further relax the minimax formulation using a given conformity score and a hypothesis class, where we also analyze the impact of this relaxation on conditional coverage validity and length optimality.

## 3.1 The Equivalent Minimax Formulation: A Duality Perspective

A natural way to think about the Primary Problem2.2 is to look at the dual formulation. Let us define,

$$g_\alpha(f, C) = \mathbb{E}_{X,Y}\left[f(X)\left\{\mathbf{1}[Y \in C(X)] - (1 - \alpha)\right\}\right] - \mathbb{E}_X\left[\text{len}(C(X))\right], \qquad (3)$$

Note that the first term in $g_\alpha(f, C)$ corresponds to coverage w.r.t. to $f$ and the second term corresponds to length. One can easily check that $g_\alpha(f, C)$ is equivalent[1]to the Lagrangian for the Primary Problem (given that $\mathcal{F}$ is an affine finite-dimensional class). The dual problem can be written as:

> **Minimax Problem:**
> $$\underset{f \in \mathcal{F}}{\text{Minimize}} \ \underset{C(x)}{\text{Maximize}} \ g_\alpha(f, C)$$

**Proposition 3.1 (Strong Duality)** *Assuming $\mathcal{D}_{Y|X}$ is continuous, the Primary Problem and the Minimax Problem are equivalent. Let $(f^*, C^*(x))$ be an optimal solution of the Minimax Problem. Then, $C^*$ is also the optimal solution of the Primary Problem. Furthermore, $C^*$ has the following form:*

$$C^*(x) = \{y \in \mathcal{Y} \mid f^*(x)p(y|x) \geq 1\} \qquad (4)$$

**Example 3.2 (Marginal case)** *For the marginal case, where $\mathcal{F} = \{x \mapsto c \mid \text{for every } c \in \mathbb{R}\}$, we have $C^*(x) = \{y \in \mathcal{Y} \mid c^*p(y|x) \geq 1\}$. In other words, the minimal length marginally valid prediction set is of the form of a specific level set of $p(y|x)$. Note that the value of $c^*$ can be found using the fact that the marginal coverage should be $1 - \alpha$.*

Simply put, Proposition 3.1 states that the answer to the Primary Problem corresponds to specific level sets of the distributions $p(y|x)$. Further, this optimal level set has an equivalent minimax formulation. We will see in section 4 how a relaxation of this minimax can be used to derive a finite sample algorithm. Next, we will discuss the roles of the inner maximization and the outer minimization.

**The Inner Maximization.** For a function $f : \mathcal{X} \to \mathbb{R}$ we define the level sets corresponding to $f$ as

$$C_f(x) = \mathbf{1}[f(x)p(y|x) \geq 1]. \qquad (5)$$

From Proposition 3.1, the optimal prediction sets have the form $C_{f^*}$ for some $f^* \in \mathcal{F}$. The following proposition, shows that for a fixed $f$ the outcome of the inner maximization *is exactly the level set $C_f$.*

**Proposition 3.3 (Variational representation)** *For any $f \in \mathcal{F}$, and $\alpha \in [0, 1]$ we have,*

$$C_f = \underset{C(x)}{argmax} \ g_\alpha(f, C), \qquad (6)$$

**The Outer Minimization.** So far, we know that for every $f$, the inner maximization chooses the level set $C_f$ given in (5). The next question is which of these level sets will be chosen by the outer minimization? The answer is simple: The one that is conditionally valid with respect to the class $\mathcal{F}$. I.e., the outer minimization chooses a $f^*$ such that $C_{f^*}$ has correct conditional coverage w.r.t. to $\mathcal{F}$.

**Lemma 3.4** *Let $f \in \mathcal{F}$. Recall that $\mathcal{F}$ is an affine class of functions. This means for every $\tilde{f} \in \mathcal{F}$ and $\varepsilon \in \mathbb{R}$, $f + \varepsilon\tilde{f} \in \mathcal{F}$. We have for every $\tilde{f} \in \mathcal{F}$,*

$$\frac{d}{d\varepsilon}g_\alpha(f + \varepsilon\tilde{f}, C_{f+\varepsilon\tilde{f}})\bigg|_{\varepsilon=0} = \mathbb{E}\left[\tilde{f}(X)\left\{\mathbf{1}[Y \in C_f(X)] - (1 - \alpha)\right\}\right].$$

At the optimum solution $f^*$ of the outer minimization, all the directional derivatives are zero (since $f^*$ is a stationary point). Hence, using the lemma, $C_{f^*}$ satisfies coverage validity on the class $\mathcal{F}$.

Let us summarize: The outer minimization ensures that we navigate the space of conditionally valid prediction sets while the inner maximization finds the most length efficient among them.

## 3.2 Relaxed Minimax Formulation using Structured Prediction Sets

Taking a closer look at the Minimax formulation 3.1, the inner maximization is over the space of all the possible prediction sets, which is an overly complex set. We need to relax this maximization due to two reasons: (1) Eventually, we would like to solve the finite-sample version of this problem

---

[1]The definition of $g_\alpha(f, C)$ is the negative of the conventional Lagrangian.

as in practice we have finite-size calibration data. Having an infinitely complex space of prediction sets simply lead to overfitting. Consequently, we need to restrict the maximization over a class of prediction sets of finite complexity. (2) In CP the prediction sets are often constructed by a carefully designed conformity score. For instance, the common practice in CP is to structure the prediction sets by threshold the conformity score. In light of these, we will need to relax the Minimax formulation.

**Structured Prediction Sets:** Given the above considerations, we will restrict the domain of inner maximization in the most natural way. Aligned with the established methods in the CP literature, we look at the prediction sets that are described by thresholding a conformity measure. Let $S(x, y) : \mathcal{X} \times \mathcal{Y} \to \mathbb{R}$ be a given conformity score. Consider a function $h : \mathcal{X} \to \mathbb{R}$. We will consider structured prediction sets of the form

$$C_h^S = \{y \in \mathcal{Y} \mid S(x, y) \leq h(x)\}. \tag{7}$$

For the ease of notation, we use $g_\alpha(f, h)$ instead of $g_\alpha(f, C_h^S)$. We then relax the Minimax Problem formulation 3.1. Let $\mathcal{H}$ be a class of real valued functions on the set $\mathcal{X}$. For now, $\mathcal{H}$ could be any class – e.g. it could be the class of all the real-valued functions on $\mathcal{X}$ (with infinite complexity) or it could be a class with bounded complexity (e.g. a neural network parameterized by its weights).

---

**Relaxed Minimax Problem:**
$$\underset{f \in \mathcal{F}}{\text{Minimize}} \, \underset{h \in \mathcal{H}}{\text{Maximize}} \, g_\alpha(f, h).$$

---

Here, the relaxation is that the inner maximization is over prediction sets of form (7) where $h \in \mathcal{H}$.

### 3.3 Theoretical Guarantees for the Relaxed Minimax Problem

A natural question to ask now is whether this relaxation preserves valid coverage guarantees or length optimality. Clearly, the answer to this question depends on the choice of the conformity measure $S$ and the class $\mathcal{H}$. Let us start by considering the structure of the optimal prediction sets given in (4); it is easy to see that the optimal sets have the form

$$C^*(x) = \left\{ y \in \mathcal{Y} \, \middle| \, \frac{1}{p(y|x)} \leq f^*(x) \right\}, \tag{8}$$

for some function $f^* : \mathcal{X} \to \mathbb{R}$.[2] Comparing (8) with (7) makes it clear that, unless the score $S(x, y)$ is aligned with $1/p(y|x)$, the structured sets $C_h^S$ may not be rich enough to capture the level sets in (8). That is to say, once we construct our prediction sets using the conformity score $S$ – i.e. using $(x, S(x, y))$ instead of $(x, y)$ – there can be a fundamental gap to the best achievable length. That is to say the level sets of $S(x, y)$ can be fundamentally different from the level sets of $p(y|x)$.

Given the above considerations, we ask: what is the appropriate notion of length-optimality that can be achieved by solving the Relaxed Minimax Problem? A natural answer is the "optimal length" that can be achieved over all the structured prediction sets $C_h^S$, $h \in \mathcal{H}$. Accordingly, we introduce the Relaxed Primary Problem which is the dual form of the Relaxed Minimax Problem:

---

**Relaxed Primary Problem:**
$$\underset{h \in \mathcal{H}}{\text{Minimize}} \quad \mathbb{E}\left[\text{len}(C_h^S(X))\right]$$
$$\text{subject to} \quad \mathbb{E}\left[f(X)\left\{\mathbf{1}[Y \in C_h^S(X)] - (1-\alpha)\right\}\right] = 0, \quad \forall f \in \mathcal{F}$$

---

To study the connection between these two problems, one would ideally want to show that the Relaxed Minimax Problem is equivalent to the Relaxed Primary Problem, similar to the strong duality relation we showed in Proposition 3.1 between the Minimax Problem 3.1 and the Primary Problem 2.2.

At a high level, we demonstrate that strong duality holds when $\mathcal{H}$ is "sufficiently complex". Specifically, we first let $\mathcal{H}$ be the class of all (measurable) functions from $\mathcal{X}$ to $\mathbb{R}$, and show that the Relaxed Minimax Problem and the Relaxed Primal Problem achieve the same optimal value—i.e., strong duality. Hence, if the Relaxed Primal Problem admits an optimal solution $h^*$ then the Relaxed Minimax

---
[2]We are assuming $\frac{1}{0} = +\infty$.

Problem has an optimal solution of the form $(f^*, h^*)$. Next, we consider a bounded-complexity class $\mathcal{H}$. We show that under a realizability assumption, i.e., $h^* \in \mathcal{H}$, strong duality still holds.

We denote the optimal value of the Relaxed Minimax Problem by OPT and the optimal value of the Relaxed Primary Problem by $L^*$. One can interpret $L^*$ as the smallest possible length over all the structured prediction sets that are conditionally valid with respect to $\mathcal{F}$. In this context, strong duality means $L^* = -\text{OPT}$, as our definition of $g_\alpha(f, h)$ differs from the conventional definition of Lagrangian by a negative sign, as we wanted to stay consistent with the literature on level set estimation (e.g. see [46]). Let us now state a technical assumption needed to develop our theory.

**Assumption 1** *Recall that the class $\mathcal{F}$ is defined as $\mathcal{F} = \{\langle \boldsymbol{\beta}, \Phi(x) \rangle | \boldsymbol{\beta} \in \mathbb{R}^d\}$, where $\Phi : \mathcal{X} \to \mathbb{R}^d$ is a predefined function (a.k.a. the finite-dimensional basis). Let $\Phi(x) = [\phi_1(x), \cdots, \phi_d(x)]$. We assume each $\phi_i(x)$ takes countably many values.*

Assumption 1 is mainly a technical assumption that helps to obtain a more simplified result. Assumption 1 holds in both marginal and group-conditional settings (as in these cases $\phi_i$'s take their value in $\{0, 1\}$). We will provide a more general but weaker result without assumption 1 in Appendix F.

**Theorem 3.5** *Let us assume $\mathcal{D}_{S|X}$ and $\mathcal{D}_X$ are continuous. Let $\mathcal{F}$ be a bounded finite-dimensional affine class of covariate shifts and $\mathcal{H}$ be the class of all (measurable) functions from $\mathcal{X}$ to $\mathbb{R}$. Under assumption 1, the strong duality holds between the Relaxed Primary Problem and the Relaxed Minimax Problem. In other words,*

$$\underset{f \in \mathcal{F}}{\text{Minimize}} \ \underset{h \in \mathcal{H}}{\text{Maximize}} \ g_\alpha(f, h) = \underset{h \in \mathcal{H}}{\text{Maximize}} \ \underset{f \in \mathcal{F}}{\text{Minimize}} \ g_\alpha(f, h),$$

*and the optimal values of the two problems coincide ($L^* = -\text{OPT}$).*

*If the Relaxed Primary Problem (i.e., the right hand side of the above equation) attains it's optimal value at $h^* \in \mathcal{H}$, then there exists $f^* \in \mathcal{F}$ such that $(f^*, h^*)$ is an optimal solution to the Relaxed Minimax Problem. Otherwise, let $f^*$ denote the optimal solution to the outer minimization of the Relaxed Minimax Problem. Then for every $\varepsilon > 0$, there exists $h^* \in \mathcal{H}$ such that,*

*(i) $|g_\alpha(f^*, h^*) - \text{OPT}| \leq \varepsilon$.*

*(ii) $h^*$ is conditionally valid; i.e., for every $f \in \mathcal{F}$ we have,*

$$\mathbb{E}\left[ f(X) \left\{ \mathbf{1}[Y \in C_{h^*}^S(X)] - (1 - \alpha) \right\} \right] = 0.$$

*(iii) $\mathbb{E}_X \text{len}(C_{h^*}^S(X)) \leq L^* + \varepsilon$.*

Put it simply, Theorem 3.5 says for any $\varepsilon > 0$, there is an $\varepsilon$-close optimal solution of the Relaxed Minimax Problem (statement (i)) such that it is a feasible solution of the Relaxed Primary Problem with exact conditional coverage (statement (ii)), and it achieves only an $\varepsilon$-larger prediction set length than the smallest possible, which is the solution of Relaxed Minimax Problem (statement (iii)).

**Bounded complexity class $\mathcal{H}$:** The following realizability condition paves the way to generalize the strong duality results to the case where $\mathcal{H}$ is a class of functions with bounded complexity.

**Definition 3.6 (Realizability)** *Let $\mathcal{H}$ be a class of real valued functions defined on $\mathcal{X}$. We say $\mathcal{H}$ is Realizable if there exists $h^* \in \mathcal{H}$ such that $h^*$ is an optimal solution of the Relaxed Primary Problem and the Relaxed Minimax Problem.*

**Proposition 3.7** *Let $\mathcal{H} = \{h_\theta \mid \theta \in \Theta\}$, be a realizable class of functions where $\Theta$ is a compact set. Then the Relaxed Minimax Problem and the Relaxed Primary Problem are equivalent. In other words, strong duality holds and the min and max are interchangeable in the Relaxed Minimax Problem.*

## 4 Finite Sample Setting: The Main Algorithm

In this section, we present and analyze our main algorithm. Assume we have access to $n$ independent and identically distributed (i.i.d.) calibration data $\{(x_i, y_i)\}_{i=1}^n$ from the distribution $\mathcal{D}$. Let $S(x, y) : \mathcal{X} \times \mathcal{Y} \to \mathbb{R}$ represents a given conformity score. We consider a bounded capacity class of functions $\mathcal{H}$–e.g. a neural network parameterized by its weights. Additionally, we consider a given finite-dimensional affine class of functions $\mathcal{F}$ that represents the level of conditional validity to be achieved.

**Unbiased Estimation.** From (3), the objective $g_\alpha(f, h)$ of the Relaxed Minimax Problem 3.2 admits a straightforward unbiased estimator using $n$ i.i.d. samples $\{(x_i, y_i)\}_{i=1}^n$:

$$g_{\alpha,n}(f, h) = \frac{1}{n} \sum_{i=1}^n \left[ f(x_i) \left\{ \mathbf{1}[S(x_i, y_i) \leq h(x_i)] - (1 - \alpha) \right\} \right] - \frac{1}{n} \sum_{i=1}^n \int_{\mathcal{Y}} \mathbf{1}[S(x_i, y) \leq h(x_i)] dy.$$

**Smoothing.** The objective $g_{\alpha,n}(f,h)$ is non-smooth as it involves indicator functions. To make it differentiable, we will need to smooth the objective. A common way to smooth the indicator function $\mathbf{1}[a < b]$ is via Gaussian smoothing. Consider the Gaussian error function, $\mathrm{erf}(x) = \frac{2}{\sqrt{\pi}} \int_0^x e^{-t^2} \, dt$, and define the smoothed indicator function as: $\tilde{\mathbf{1}}(a,b) = \frac{1}{2}\left(1 + \mathrm{erf}\left(\frac{a-b}{\sqrt{2}\sigma}\right)\right)$, where $\sigma$ controls the smoothness of the transition between 0 to 1. The value of $\sigma$ is often chosen to be small, and when it approaches zero we retrieve the indicator function. The smoothed version of our objective is

$$\tilde{g}_{\alpha,n}(f,h) = \frac{1}{n}\sum_{i=1}^n \left[f(x_i)\left\{\tilde{\mathbf{1}}(S(x_i,y_i),h(x_i)) - (1-\alpha)\right\}\right] - \frac{1}{n}\sum_{i=1}^n \int_{\mathcal{Y}} \tilde{\mathbf{1}}(S(x_i,y),h(x_i))dy.$$

Given this smoothed objective, our main algorithm is presented in Algorithm 1.

---

**Algorithm 1** Conformal Prediction with Length-Optimization (CPL)

---

1: **Input:** Miscoverage level $\alpha$, conditional coverage requirements $\mathcal{F}$, conformity score $S$, class $\mathcal{H}$.
2: **Objective:**
$$\underset{f\in\mathcal{F}}{\text{Minimize}} \ \underset{h\in\mathcal{H}}{\text{Maximize}} \ \tilde{g}_{\alpha,n}(f,h).$$
3: **while** not converged **do**
4:     Perform a few iterations (e.g. SGD steps) on $h$ to maximize $\tilde{g}_{\alpha,n}(f,h)$
5:     Perform a few iterations (e.g. SGD steps) on $f$ to minimize $\tilde{g}_{\alpha,n}(f,h)$
6: **end while**
7: $f_{\mathrm{CPL}}^* \leftarrow f, \quad h_{\mathrm{CPL}}^* \leftarrow h$
8: Let $C_{\mathrm{CPL}}^*(x) = \{y \in \mathcal{Y} \mid S(x,y) \leq h_{\mathrm{CPL}}^*(x)\}$.

---

**Remark 4.1** *Oftentimes in practice, the machine learning models, such as Neural Networks, are parametric ($h_\theta$, where $\theta$ is the set of parameters). In that case, performing the iteration on $h$ can be implemented by updating $\theta$.*

**Remark 4.2** *Recalling the definition of $\mathcal{F} = \{\langle \boldsymbol{\beta}, \Phi(x)\rangle | \boldsymbol{\beta} \in \mathbb{R}^d\}$, each $f \in \mathcal{F}$ can be represented by $f(x) = \langle \boldsymbol{\beta}, \Phi(x)\rangle \equiv f_{\boldsymbol{\beta}}(x)$. Hence, performing the iteration on $f$ can be implements by updating $\boldsymbol{\beta} \in \mathbb{R}^d$.*

### 4.1 Finite Sample Guarantees

In this section we provide finite sample guarantees for the conditional coverage of the prediction sets constructed by CPL using the covering number of class $\mathcal{H}$ (denoted by $\mathcal{N}$). We analyse the properties of stationary points of CPL. This is not a very restrictive choice as it is well known in the optimization literature that the converging points of gradient descent ascent methods (like CPL) are the stationary points of the minimax problem (e.g. look at [47]). We now state the required technical assumption.

**Definition 4.3** *A distribution $\mathcal{P}$, is called L-lipschitz if we have for every real valued numbers $q \leq q'$,*

$$\Pr_{X\sim\mathcal{P}}(X \leq q') - \Pr_{X\sim\mathcal{P}}(X \leq q) \leq L(q'-q).$$

**Assumption 2** *The conditional distribution $\mathcal{D}_{S|X}$ is L-Lipschitz, almost surely with respect to $\mathcal{D}_X$.*

Assumption 2 is often needed for concentration guarantees in CP literature [4, 11, 48, 49]. Consider CPL with smoothed function $\tilde{g}_{\alpha,n}$ with smoothing parameter $\sigma = \frac{1}{\sqrt{n}}$. Let $(f_{\mathrm{CPL}}^*, h_{\mathrm{CPL}}^*)$ denote a stationary point reached by Algorithm 1. Also, the corresponding prediction sets are given as $C_{\mathrm{CPL}}^*(x) = \{y \in \mathcal{Y} \mid S(x,y) \leq h_{\mathrm{CPL}}^*(x)\}$.

**Theorem 4.4 (Conditional coverage validity)** *Under assumption 2, let $(f_{\mathrm{CPL}}^*, h_{\mathrm{CPL}}^*)$ denote a stationary point reached by Algorithm 1 (if exists) and $C_{\mathrm{CPL}}^*(x)$ its corresponding prediction sets. Then with probability $1-\delta$ we have,*

$$\left|\mathbb{E}\left[f_{\boldsymbol{\beta}}(X)\left\{\mathbf{1}[Y \in C_{\mathrm{CPL}}^*(X)] - (1-\alpha)\right\}\right]\right| \leq \frac{c_1\sqrt{\ln\left(\frac{2d\mathcal{N}(\mathcal{H}, d_\infty, \frac{1}{n})}{\delta}\right)} + c_2}{\sqrt{n}}, \quad \forall f_{\boldsymbol{\beta}} \in \mathcal{F},$$

*where $c_1 = \sqrt{2}B\|\boldsymbol{\beta}\|_1$, $c_2 = \|\boldsymbol{\beta}\|_1 BL\sqrt{\frac{\pi}{2}} + \|\boldsymbol{\beta}\|_1\frac{2B}{\sqrt{2\pi}}$, $B = \max_i \sup_{x\in\mathcal{X}} \Phi_i(x)$, and $\Phi$ is the basis for $\mathcal{F}$ (see Section 2.1).*

PAC-style guarantees of the order $O(\frac{1}{\sqrt{n}})$ are generally optimal and common in CP literature [1, 4, 50]. Look at Appendix C for further discussions and interpretations.

# 5 Experiments

In this section, we empirically examine the length performance of CPL compared to state-of-the-art (SOTA) baselines under varying levels of conditional validity requirements. This section is organized into three parts, each dedicated to setups that demand a specific level of conditional validity. Throughout this section, we demonstrate across different data modalities and experimental settings that CPL provides significantly tighter prediction sets (i.e. smaller length) that are conditionally valid. Additionally, we have provided a Python notebook with a step-by-step efficient implementation of our algorithm, which can be accessed at the following link: https://github.com/shayankiyani98/CP.

## 5.1 Part I: Marginal Coverage for Multiple Choice Question Answering

We use multiple-choice question answering datasets, including TruthfulQA [51], MMLU [52], OpenBookQA [53], PIQA[54], and BigBench [55]. The task is to quantify the uncertainty of Llama 2 [56] and create prediction sets using this model. We follow a procedure similar to the one proposed by [33] described below.

For each dataset, we tackle the task of multiple-choice question answering. We pass the question to Llama 2 using a simple and fixed prompt engineering approach throughout the experiment:

```
This is a 4-choice question that you should answer:
{question} The correct answer to this question is:
```

We then look at the logits of the first output token for the options A, B, C, and D. By applying a softmax function, we obtain a probability vector. The conformity score is defined as $(1 - \text{probability of the correct option})$, which is similar to $(1 - f(x)_y)$ for classification.

Our method is implemented using $\mathcal{H}$ as a linear head on top of a pre-trained GPT-2 [57]. GPT-2 has 768-dimensional hidden layers, so the optimization for the inner maximization involves a 768-dimensional linear map from GPT-2's last hidden layer representations to a real-valued scalar. We also implemented the method of [33], which directly applies split conformal method on the scores.

Figures 3a shows the performance of our method (CPL) compared to the baseline. CPL achieves significantly smaller set sizes while ensuring proper 90% coverage.

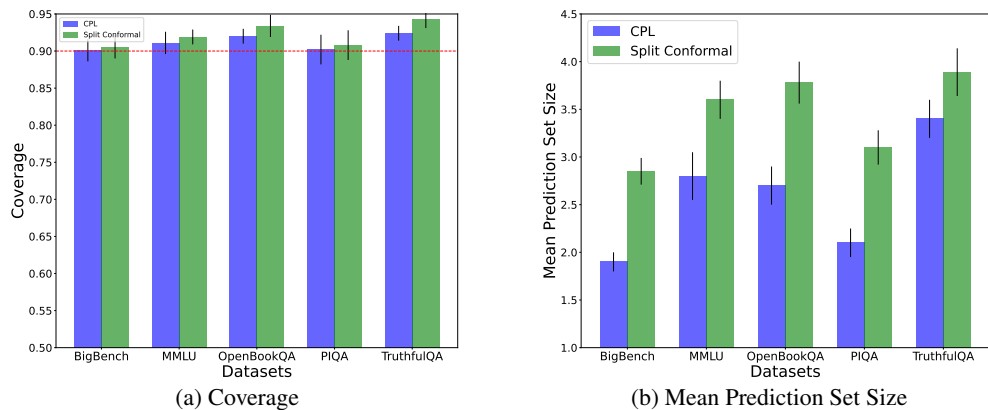

(a) Coverage                                            (b) Mean Prediction Set Size

Figure 3: Left-hand-side plot shows coverage and right-hand-side shows mean prediction set size.

## 5.2 Part II: Group-Conditional Coverage for Synthetic Regression

We use a synthetic regression task as in [4] to compare CPL with BatchGCP [4]. The covariate $X = (X_1, \cdots, X_{100})$ is a vector in $\mathbb{R}^{100}$. The first ten coordinates are independent uniform binary variables, and the remaining 90 coordinates are i.i.d. Gaussian variables. The label $y$ is generated as $y = \langle \theta, X \rangle + \epsilon_X$ where $\epsilon_X$ is a zero-mean Gaussian with variance $\sigma_X^2 = 1 + \sum_{i=1}^{10} iX_i + (40 \cdot \mathbf{1}\left[\sum_{i=11}^{100} X_i \geq 0\right] - 20)$. We generate 150K training samples, 50K calibration data points, and 50K test data points. We evaluate all algorithms over 100 trials and report average performance.

We define 20 overlapping groups based on ten binary components of $X$. For each $i$ in 1 to 10, Group $2i - 1$ corresponds to $X_i = 0$ and Group $2i$ to $X_i = 1$. BatchGCP and CPL are implemented to provide group-conditional coverage for these groups. We use a 2-hidden-layer NN with layers of

20 and 10 neurons for the inner maximization. We also include the Split Conformal method and an optimal oracle baseline, calculated numerically using the optimal formulation of Proposition 3.1.

As seen in Figure 4, the Split Conformal under covers in high-variance groups and over covers in low-variance groups. CPL and BatchGCP provide near-perfect coverage in all groups, matching the Optimal Oracle. The mean interval plot shows that BatchGCP performs similarly to Split Conformal, while CPL significantly reduces the average length, nearing the Optimal Oracle solution.

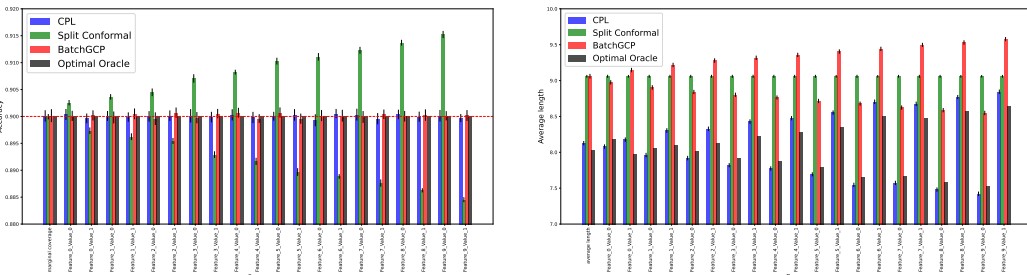

Figure 4: Left-hand-side plot shows coverage and right-hand-side shows mean interval length.

## 5.3 Part III: Coverage w.r.t. a General Class of Covariate Shifts for Image Classification

We conduct experiments on the RxRx1 dataset from the WILDS repository, which consists of cell images for predicting one of 1,339 genetic treatments across 51 experiments, each exhibiting covariate shifts. Our objective is to construct prediction sets that retain valid coverage across these shifts. We compare two methods: CPL and the Conditional Calibration algorithm introduced by [8], which employs quantile regression to achieve valid conditional coverage under covariate shifts.

To implement these methods, we follow the approach in [8], where covariate shifts are defined by partitioning the calibration data. A multinomial linear regression model with $\ell_2$ regularization is then trained on the first half of the calibration data to predict which experiment each image belongs to, and the features learned by this regression serve as the basis of the covariate shift class.

For the genetic treatment prediction task, we use a pre-trained ResNet50 model (trained by [58]), $f(x)$, trained on data from 37 experiments. The remaining 14 experiments are split into separate calibration and test sets. To handle the inner optimization of our method, we train a linear classifier on the final-layer representations of the pre-trained ResNet50. We then compute conformity scores as follows: for each image $x$, let $f^i(x)_{i=1}^{1339}$ represent the output of the ResNet50 for each treatment class. We apply temperature scaling and a softmax function to convert these outputs into probability estimates, $\pi^i(x) := \exp(Tf_i(x))/(\sum_j \exp(Tf_j(x)))$, where $T$ is the temperature parameter. The conformity score $S(x, y)$ is calculated as the sum of $\pi_i(x)$ over treatments for which $\pi_i(x) > \pi_y$.

Figure 5 compares CPL, Conditional Calibration, and Split Conformal methods. While both CPL and Conditional Calibration maintain valid coverage, CPL achieves significantly smaller average prediction set sizes by focusing on feature learning without sacrificing conditional validity.

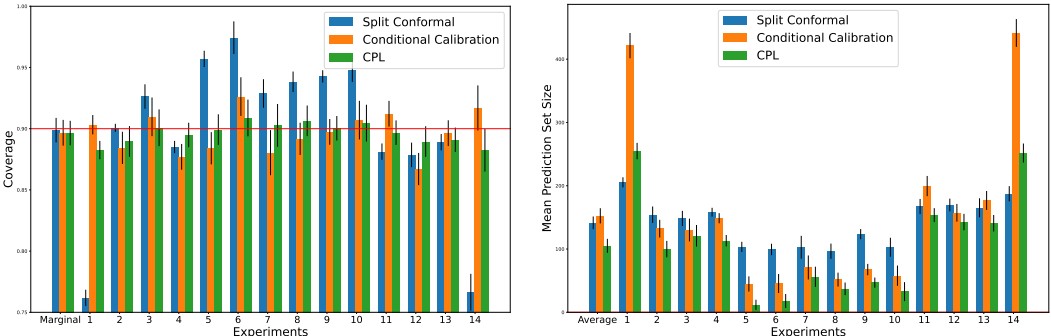

Figure 5: Left-hand-side plot shows coverage and right-hand-side shows mean prediction set size. The reported values are averaged over 20 different splits of calibration data.

## Acknowledgments

This work is supported by the NSF Institute for CORE Emerging Methods in Data Science (EnCORE). The authors wish to thank Luiz Chamon for helpful discussions.

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

# A   Outline

- Section B provides more discussions on the existing related works.
- Appendix C is dedicated on some interpretations of the finite sample guarantees of Section 4.
- Appendix D lists the all the Remarks that we moved to Appendix from the main body due to space limit.
- Appendix E provides the Lemmas, including their proofs, that are necessary for the development of out theoretical framework.
- Section F provides the proofs of all the statements proposed in Section 3 along with an extra Theorem.
- Section G provides the proofs of all the statements proposed in Section 4.
- Section H provides an extensive experimental setting to showcase the ability of CPL to improve length efficiency in real-world regression scenarios.
- Section I provides an extra experiment setup that showcases the possibility of applying CPL on top of a CIFAR-10 classifier that is trained by conformal training.
- Section J provides references to all the 11 regression datasets used in experiment setup of Section H.

# B   Further Related Works

We now review some of the remaining prior works that are relevant.

**Designing better conformity scores:** There is a growing body of work on designing better conformity scores to improve conditional validity and/or length efficiency [22, 35–41, 59] (see [60] for more references). As shown in Fig. 1, our framework applies post selection of the score, and hence can use any of the conformity scores designed in the literature (see Section 5 for experiments with different scores).

**Level set estimation.** As we will see, a key ingredient to our framework is to study the problem of CP through the lens of level set estimation. Level set estimation has been extensively studied in the literature of statistics [46, 61–63], and it was introduced to CP community by [64] and further studied by [23]. Our framework expands and builds upon the connection of level set estimation and CP. In particular, comparing to [23, 64], we take three significant steps. (i) We propose to utilize the covariate, $X$, to directly *optimize length via an adaptive threshold*. This results in major improvements in length efficiency in practice. (ii) We go beyond marginal coverage to different levels of conditional validity. (iii) Finally, unlike [64], we do not need to directly estimate any marginal\conditional density function. Level sets and their approximations appear naturally in our optimization framework.

**Conformal training:** Several works have focused on directly optimizing the score function to enhance the length efficiency of conformal prediction pipelines [24, 34, 65]. In particular, [65] introduced conformal training, where a (marginal) calibration procedure is fixed, and derivatives of the prediction set size with respect to the model parameters are taken. This enables the model to be optimized not for predictive accuracy, as is typical, but for producing smaller prediction sets. However, a key distinction separates our approach from the broader idea of conformal training. To illustrate this difference, consider the simpler case where we aim only to improve length efficiency under marginal coverage. Let $S_\theta(x, y) : \mathcal{X} \times \mathcal{Y} \to \mathbb{R}$ be a parameterized conformity score (e.g., where $\theta$ represents the parameters of a neural network). Conformal training optimizes $\theta$ to improve the length efficiency of prediction sets of the form $S_\theta(x, y) \leq q$, where $q$ is a fixed threshold. In contrast, we keep $\theta$ fixed and optimize an adaptive threshold $h(x) : \mathcal{X} \to \mathbb{R}$ to improve the length efficiency of prediction sets of the form $S_\theta(x, y) \leq h(x)$. That is to say, our method can also be applied on top of a score trained by conformal training. In appendix I, we show that it is possible to optimize length even further by applying our method on top of the conformal training introduced by [65]. It's important to note that our use of covariate-adaptive thresholds is specifically for length optimization, distinct from the use in addressing conditional coverage in prior literature. Beyond the theoretical framework, our perspective allows for length optimization in various real-world applications. In practice, access to a

model's internal parameters may be restricted due to privacy, security, or policy concerns. In section 5.1, we demonstrate how to construct efficient prediction sets for large language models (LLMs) without optimizing their internal parameters. Additionally, in uncertainty quantification, conformal prediction sets often serve to assess the behavior of a given black-box model. In these cases, the statistician's role is to analyze the model's behavior rather than further optimizing it, which could degrade performance on nominal or out-of-distribution tasks. Additionally, in a concurrent work to ours, [34] extended conformal training idea to a more general case, allowing the score function to be optimized while targeting conditional coverage in the calibration procedure. They develop a novel technique to differentiate through the conditional conformal algorithm introduced by [8]. However, while both approaches utilize similar relaxations for targeting conditional coverage, our work diverges in the underlying objective. Whereas [34] focuses on optimizing the score function, we emphasize optimizing the threshold leaving the scores untouched. This leads to two distinct theoretical and algorithmic developments.

## C   Further Discussions on Finite Sample Result

**Theorem C.1 (Conditional coverage validity)** *Under assumption 2, let $(f^*_{\text{CPL}}, h^*_{\text{CPL}})$ denote a stationary point reached by Algorithm 1 (if exists) and $C^*_{\text{CPL}}(x)$ its corresponding prediction sets. Then with probability $1 - \delta$ we have,*

$$\left| \mathbb{E}\left[ f_{\boldsymbol{\beta}}(X) \left\{ \mathbf{1}[Y \in C^*_{\text{CPL}}(X)] - (1 - \alpha) \right\} \right] \right| \leq \frac{c_1 \sqrt{\ln\left( \frac{2d\mathcal{N}(\mathcal{H}, d_\infty, \frac{1}{n})}{\delta} \right)} + c_2}{\sqrt{n}}, \quad \forall f_{\boldsymbol{\beta}} \in \mathcal{F},$$

*where $c_1 = \sqrt{2}B\|\boldsymbol{\beta}\|_1$, $c_2 = \|\boldsymbol{\beta}\|_1 BL\sqrt{\frac{\pi}{2}} + \|\boldsymbol{\beta}\|_1 \frac{2B}{\sqrt{2\pi}}$, $B = \max_i \sup_{x \in \mathcal{X}} \Phi_i(x)$, and $\Phi$ is the basis for $\mathcal{F}$ (see Sec. 2.1).*

We now provide three Corollaries to this Theorem.

**The case of marginal validity:** For this case we have to pick $\mathcal{F} = \{x \mapsto c \mid c \in \mathbb{R}\}$, i.e. constant functions. In this case one can see $B = 1$, so we have the following result.

**Corollary C.2 (Finite sample marginal validity)** *Under assumption 2, let $(f^*_{\text{CPL}}, h^*_{\text{CPL}})$ denote a stationary point reached by Algorithm 1 (if exists) and $C^*_{\text{CPL}}(x)$ its corresponding prediction sets. Then with probability $1 - \delta$ we have,*

$$\left| \Pr(Y \in C^*_{\text{CPL}}(X)) - (1 - \alpha) \right| \leq \frac{c_1 \sqrt{\ln\left( \frac{2\mathcal{N}(\mathcal{H}, d_\infty, \frac{1}{n})}{\delta} \right)} + c_2}{\sqrt{n}},$$

*where $c_1 = \sqrt{2}$, $c_2 = L\sqrt{\frac{\pi}{2}} + \frac{2}{\sqrt{2\pi}}$.*

**The case of group-conditional validity:** Let $G_1, \cdots, G_m$ be a collection of groups: each group is a subset of covariate space, i.e. $G_i \in \mathcal{X}$. These groups can be fully arbitrary and highly overlapping. Define $f_i(x) = \mathbf{1}[x \in G_i]$ for every $i \in [1, m]$. We can then run CPL using the class of covariate shifts $\mathcal{F} = \{\sum_{i=1}^m \beta_i f_i(x) \mid \beta_i \in \mathbb{R} \text{ for every } i \in [1, \cdots, m]\}_{i=1}^m$ and obtain tight prediction sets with group-conditional coverage validity. In this case one can see $B = 1$, so we have the following result.

**Corollary C.3 (Finite sample group-conditional validity)** *Under assumption 2, let $(f^*_{\text{CPL}}, h^*_{\text{CPL}})$ denote a stationary point reached by Algorithm 1 (if exists) and $C^*_{\text{CPL}}(x)$ its corresponding prediction sets. Then with probability $1 - \delta$ we have,*

$$\left| \Pr(Y \in C^*_{\text{CPL}}(X) \mid X \in G_i) - (1 - \alpha) \right| \leq \frac{c_1 \sqrt{\ln\left( \frac{2m\mathcal{N}(\mathcal{H}, d_\infty, \frac{1}{n})}{\delta} \right)} + c_2}{\sqrt{n}}, \quad \forall i \in [1, \cdots, m],$$

*where $c_1 = \sqrt{2}$, $c_2 = L\sqrt{\frac{\pi}{2}} + \frac{2}{\sqrt{2\pi}}$.*

# D  Further Remarks

**Remark D.1** *Another important special case of conditional validity with respect to an affine finite dimensional class of covariate shifts is the framework introduced by [9] which provides CP methods that guarantee a valid coverage when there is a known covariate shift between calibration data and test data. This exactly falls to our framework as the special case if we consider the class $\mathcal{F} = \{x \to cf(x) \mid \text{for every } c \in \mathbb{R}\}$, where $f$ is the known covariate shift (i.e. likelihood ratio between test and calibration data).*

**Remark D.2** *We will need to assume that the members of the class $\mathcal{H}$ are bounded functions; i.e. for every $h \in \mathcal{H}$ and $x \in \mathcal{X}$ we have $h(x) \in [0, \Gamma]$. Two points are in order. (i) This assumption is purely for the sake of theory development. In practice, one can run our algorithm with any off-the-shelf machine learning class of models. In fact, in Section 5, we run our algorithm with a variety of models including deep neural networks (Resnet 50) and show case its performance. (ii) This assumption is not very far from practice. In fact, one can satisfy this assumption by using a Sigmoid activation function (or a scaled version of it) on the output layer to satisfy this assumption. Similar assumptions has been posed in the literature [4, 11, 48, 49].*

# E  Technical Lemmas

**Lemma E.1** *Let $f(a, x) = \frac{1}{2}\left(1 + erf\left(\frac{a-x}{\sqrt{2}\sigma}\right)\right)$ be the smoothed indicator function, where $erf(x)$ is the error function, defined as $erf(x) = \frac{2}{\sqrt{\pi}}\int_0^x e^{-t^2}\, dt$, and $\sigma$ is the variance of the Gaussian kernel used for smoothing. Then $f(a, x)$ is Lipschitz continuous with respect to $x$ with a Lipschitz constant $\frac{1}{\sqrt{2\pi}\sigma}$.*

**Proof:** To prove that $f(a, x)$ is Lipschitz continuous with respect to $x$, we compute the derivative of $f(a, x)$ with respect to $x$:

$$f(a, x) = \frac{1}{2}\left(1 + \operatorname{erf}\left(\frac{a - x}{\sqrt{2}\sigma}\right)\right)$$

$$\frac{\partial}{\partial x}f(a, x) = \frac{1}{2}\frac{\partial}{\partial x}\left(1 + \operatorname{erf}\left(\frac{a - x}{\sqrt{2}\sigma}\right)\right) = \frac{1}{2}\cdot\frac{2}{\sqrt{\pi}}\cdot\frac{-1}{\sqrt{2}\sigma}e^{-\left(\frac{a-x}{\sqrt{2}\sigma}\right)^2}$$

Simplifying this, we get:

$$\frac{\partial}{\partial x}f(a, x) = -\frac{1}{\sqrt{2\pi}\sigma}e^{-\frac{(a-x)^2}{2\sigma^2}}$$

To show that this derivative is bounded, observe that:

$$\left|\frac{\partial}{\partial x}f(a, x)\right| = \left|-\frac{1}{\sqrt{2\pi}\sigma}e^{-\frac{(a-x)^2}{2\sigma^2}}\right| \leq \frac{1}{\sqrt{2\pi}\sigma}$$

Since $\frac{1}{\sqrt{2\pi}\sigma}$ is a constant, we conclude that $f(a, x)$ is Lipschitz continuous with respect to $x$ with Lipschitz constant $\frac{1}{\sqrt{2\pi}\sigma}$.

**Lemma E.2** *Let $\tilde{\mathbf{1}}[a < b]$ be the smoothed indicator function defined by*

$$\tilde{\mathbf{1}}[a < b] = \frac{1}{2}\left(1 + erf\left(\frac{a - b}{\sqrt{2}\sigma}\right)\right)$$

*where $erf(x)$ is the error function, defined as $erf(x) = \frac{2}{\sqrt{\pi}}\int_0^x e^{-t^2}\, dt$, and $\sigma$ is the variance of the Gaussian kernel used for smoothing. The indicator function $\mathbf{1}[a < b]$ is also defined as*

$$\mathbf{1}[a < b] = \begin{cases} 1 & \text{if } a < b \\ 0 & \text{otherwise} \end{cases}$$

*The error between the smoothed indicator function $\tilde{\mathbf{1}}[a < b]$ and the actual indicator function $\mathbf{1}[a < b]$, integrated over all $a$, is given by*

$$E = \int_{-\infty}^{\infty} \left| \tilde{\mathbf{1}}[a < b] - \mathbf{1}[a < b] \right| \, da$$

*This integral evaluates to*

$$E = \sqrt{\frac{\pi}{2}} \sigma$$

**Proof:** To compute the integral, we consider the two cases separately: $a < b$ and $a \geq b$.

For $a < b$, the indicator function $\mathbf{1}[a < b] = 1$, so the absolute difference is

$$\left| \tilde{\mathbf{1}}[a < b] - 1 \right| = \left| \frac{1}{2} \left( 1 + \text{erf}\left( \frac{a - b}{\sqrt{2}\sigma} \right) \right) - 1 \right| = \frac{1}{2} \left| \text{erf}\left( \frac{a - b}{\sqrt{2}\sigma} \right) - 1 \right|$$

For $a \geq b$, the indicator function $\mathbf{1}[a < b] = 0$, so the absolute difference is

$$\left| \tilde{\mathbf{1}}[a < b] - 0 \right| = \frac{1}{2} \left( 1 + \text{erf}\left( \frac{a - b}{\sqrt{2}\sigma} \right) \right)$$

Combining these, the integral can be written as

$$E = \int_{-\infty}^{b} \frac{1}{2} \left| \text{erf}\left( \frac{a - b}{\sqrt{2}\sigma} \right) - 1 \right| \, da + \int_{b}^{\infty} \frac{1}{2} \left( 1 + \text{erf}\left( \frac{a - b}{\sqrt{2}\sigma} \right) \right) \, da$$

We know that

$$\text{erf}(x) = 2\Phi(x\sqrt{2}) - 1$$

where $\Phi(x)$ is the CDF of the standard normal distribution.

Using symmetry properties of the error function and Gaussian integrals, we can simplify the calculations as follows:

$$\int_{-\infty}^{b} \left( \text{erf}\left( \frac{a - b}{\sqrt{2}\sigma} \right) - 1 \right) \, da = -\sqrt{\frac{\pi}{2}} \sigma$$

$$\int_{b}^{\infty} \left( 1 + \text{erf}\left( \frac{a - b}{\sqrt{2}\sigma} \right) \right) \, da = \sqrt{\frac{\pi}{2}} \sigma$$

Thus, the total error integral is:

$$E = \frac{1}{2} \left( \sqrt{\frac{\pi}{2}} \sigma + \sqrt{\frac{\pi}{2}} \sigma \right) = \frac{1}{2} \left( 2\sqrt{\frac{\pi}{2}} \sigma \right) = \sqrt{\frac{\pi}{2}} \sigma$$

**Lemma E.3** *Let $f \in \mathcal{F}$ be a fixed function such that $\sup_{x \in \mathcal{X}} f(x) \leq B$ and let us define,*

$$Z_h = \left| \frac{1}{n} \sum_{i=1}^{n} \left[ f(x_i) \left\{ \tilde{\mathbf{1}}(S(x_i, y_i), h(x_i)) - (1 - \alpha) \right\} \right] \right.$$

$$\left. - \mathbb{E}\left[ f(x) \left\{ \tilde{\mathbf{1}}(S(x, y), h(x)) - (1 - \alpha) \right\} \right] \right|$$

*The following uniform convergence holds: Fixing any $\varepsilon > 0$, we have with probability $1 - \delta$,*

$$|Z_h| \leq \frac{2B\varepsilon}{\sqrt{2\pi}\sigma} + \frac{\sqrt{2}B\sqrt{\ln\left( \frac{2\mathcal{N}(\mathcal{H}, d_{\infty}, \varepsilon)}{\delta} \right)}}{\sqrt{n}} \quad \text{for every } h \in \mathcal{H}.$$

**Proof.** Let $h_1, h_2 \in \mathcal{H}$ be two arbitrary functions. We have,

$$
\begin{aligned}
|Z_{h_1} - Z_{h_2}| &\overset{(a)}{\leq} \left| \frac{1}{n} \sum_{i=1}^{n} \left[ f(x_i) \Big\{ \tilde{\mathbf{1}}(S(x_i, y_i), h_1(x_i)) - \tilde{\mathbf{1}}(S(x_i, y_i), h_2(x_i)) \Big\} \right] \right. \\
&\qquad \left. - \mathbb{E}\left[ f(x) \Big\{ \tilde{\mathbf{1}}(S(x, y), h_1(x)) - \tilde{\mathbf{1}}(S(x, y), h_2(x)) \Big\} \right] \right| \\
&\overset{(b)}{\leq} \left| \frac{1}{n} \sum_{i=1}^{n} \left[ f(x_i) \Big\{ \tilde{\mathbf{1}}(S(x_i, y_i), h_1(x_i)) - \tilde{\mathbf{1}}(S(x_i, y_i), h_2(x_i)) \Big\} \right] \right| \\
&\qquad + \left| \mathbb{E}\left[ f(x) \Big\{ \tilde{\mathbf{1}}(S(x, y), h_1(x)) - \tilde{\mathbf{1}}(S(x, y), h_2(x)) \Big\} \right] \right| \\
&\overset{(c)}{\leq} B \left| \frac{1}{n} \sum_{i=1}^{n} \left[ \Big\{ \tilde{\mathbf{1}}(S(x_i, y_i), h_1(x_i)) - \tilde{\mathbf{1}}(S(x_i, y_i), h_2(x_i)) \Big\} \right] \right| \\
&\qquad + B \left| \mathbb{E}\left[ \Big\{ \tilde{\mathbf{1}}(S(x, y), h_1(x)) - \tilde{\mathbf{1}}(S(x, y), h_2(x)) \Big\} \right] \right| \\
&\overset{(d)}{\leq} \frac{B}{\sqrt{2\pi}\sigma} \left| \frac{1}{n} \sum_{i=1}^{n} \Big[ |h_1(x_i) - h_2(x_i)| \Big] \right| \\
&\qquad + \frac{B}{\sqrt{2\pi}\sigma} \left| \mathbb{E}\left[ |h_1(x) - h_2(x)| \right] \right| \\
&\overset{(e)}{\leq} \frac{2B}{\sqrt{2\pi}\sigma} \sup_{x \in \mathcal{X}} |h_1(x) - h_2(x)|
\end{aligned}
$$

where (a) is a triangle inequality, (b) is another triangle inequality, (c) comes from the definition of $B$, (d) follows from the Lipschitness Lemma E.1, and finally (e) is from the definition of sup.

Now let us define $d_\infty(h_1, h_2) = \sup_{x \in \mathcal{X}} |h_1(x) - h_2(x)|$. Therefore, $|Z_{h_1} - Zh_2| \leq 2Bd_\infty(h_1, h_2)$. By Hoeffding's inequality for general bounded random variables (look at chapter 2 of [66]), fixing $h \in \mathcal{H}$, we have with probability $1 - \delta$,

$$
|Z_h| \leq \frac{\sqrt{2}B\sqrt{\ln\left(\frac{2}{\delta}\right)}}{\sqrt{n}}. \tag{9}
$$

Now as a result of Lemma 5.7 of [67] (a standard covering number argument) we conclude,

$$
|Z_h| \leq \frac{2B\varepsilon}{\sqrt{2\pi}\sigma} + \frac{\sqrt{2}B\sqrt{\ln\left(\frac{2\mathcal{N}(\mathcal{H}, d_\infty, \varepsilon)}{\delta}\right)}}{\sqrt{n}} \quad \text{for every } h \in \mathcal{H}.
$$

**Lemma E.4** *The following equivalence between the three problems hold,*

*Structured Minimax Problem*

$$\equiv$$

*Convex Structured Minimax Problem*

$$\equiv$$

*−Convex Structured Primary Problem*

where by -**Convex Structured Primary Problem** we mean the optimal value of **Convex Structured Primary Problem** is equal to the negative of the optimal values of the other two problems.

*Proof.* We start by the following claim which will be proven shortly.

**Claim 1** *Convex Structured Minimax Problem is equivalent to the Structured Minimax Problem.*

*Proof.* Let us remind the objective,

$$g_\alpha(f, C) = \mathbb{E}\left[f(X)\Big\{C(X, Y) - (1 - \alpha)\Big\}\right] - \mathbb{E}\int_{\mathcal{Y}} C(X, y)dy,$$

which is linear in terms of $C$. Now fixing $f \in \mathcal{F}$, since $\mathcal{C}_{\mathcal{H}^\infty} \in \mathcal{C}_{\mathcal{H}^\infty}^{\mathrm{con}}$ we have,

$$\underset{C \in \mathcal{C}_{\mathcal{H}^\infty}^{\mathrm{con}}}{\mathrm{Maximize}}\ g_\alpha(f, C) \geq \underset{C \in \mathcal{C}_{\mathcal{H}^\infty}}{\mathrm{Maximize}}\ g_\alpha(f, C).$$

We now proceed by showing the other direction. Let $\{C_i\}_{i=1}^\infty$, where $C_i \in \mathcal{C}_{\mathcal{H}^\infty}^{\mathrm{con}}$ and $\lim_{i\to\infty} g_\alpha(f, C_i) = \underset{C \in \mathcal{C}_{\mathcal{H}^\infty}^{\mathrm{con}}}{\mathrm{Maximize}}\ g_\alpha(f, C)$ (pay attention that it is not trivial that this maximum is achievable by a single member of its domain). Now by the definition of $\mathcal{C}_{\mathcal{H}^\infty}^{\mathrm{con}}$, we know each $C_i$ is a convex combination of finitely many members of $\mathcal{C}_{\mathcal{H}^\infty}$. That is to say, for each $C_i$, there exists $\{C_{ij}\}_{j=1}^T$ where $C_{ij} \in \mathcal{C}_{\mathcal{H}^\infty}$ and,

$$C_i = \sum_{j=1}^T a_j C_{ij}(x, y), \quad \sum_{j=1}^T a_j = 1, a_j \geq 0 (\forall 1 \leq i \leq T).$$

Hence,

$$g_\alpha(f, C_i) = g_\alpha(f, \sum_{j=1}^T a_j C_{ij}(x, y)) = \sum_{j=1}^T a_j g_\alpha(f, C_{ij}) \leq \overset{T}{\underset{j=1}{\max}}\ g_\alpha(f, C_{ij})$$

Let us assume that final maximum is achieved by the index $j_i$. Therefore, for each $C_i \in \mathcal{C}_{\mathcal{H}^\infty}^{\mathrm{con}}$ there exists a $C_{ij_i} \in \mathcal{C}_{\mathcal{H}^\infty}$ such that $g_\alpha(f, C_i) \leq g_\alpha(f, C_{ij_i})$. Hence we have,

$$\underset{C \in \mathcal{C}_{\mathcal{H}^\infty}^{\mathrm{con}}}{\mathrm{Maximize}}\ g_\alpha(f, C) = \lim_{i\to\infty} g_\alpha(f, C_i) \leq \lim_{i\to\infty} g_\alpha(f, C_{ij_i}) \leq \underset{C \in \mathcal{C}_{\mathcal{H}^\infty}}{\mathrm{Maximize}}\ g_\alpha(f, C)$$

Putting everything together, for every $f \in \mathcal{F}$ we have,

$$\underset{C \in \mathcal{C}_{\mathcal{H}^\infty}^{\mathrm{con}}}{\mathrm{Maximize}}\ g_\alpha(f, C) = \underset{C \in \mathcal{C}_{\mathcal{H}^\infty}}{\mathrm{Maximize}}\ g_\alpha(f, C),$$

which concludes the claim. □

Now Let us define the Convex Structured Primary Problem.

---

**Convex Structured Primary Problem:**

$$\underset{C \in \mathcal{C}_{\mathcal{H}^\infty}^{\mathrm{con}}}{\mathrm{Minimize}} \quad \mathbb{E}\left[\mathrm{len}(C(X))\right]$$

$$\mathrm{subject\ to} \quad \mathbb{E}\left[f(X)\Big\{C(X, Y) - (1 - \alpha)\Big\}\right] = 0, \quad \forall f \in \mathcal{F}$$

---

As the Convex Structured Primary Problem is a linear minimization over a convex set, $\mathcal{C}_{\mathcal{H}^\infty}^{\mathrm{con}}$, with finitely many linear constraints, strong duality holds for this problem (See Theorem 1, Section 8.3 of [68]; Also see Problem 7 in Chapter 8 of the same reference), which means Convex Structured Primary Problem is equivalent to Convex Structured Minimax Problem (Here we are using the fact that $\mathcal{D}_{S|X}$ is continuous hence the Convex Structured Primary Problem is feasible). Now putting everything together we have,



**Structured Minimax Problem**

$\equiv$

**Convex Structured Minimax Problem**

$\equiv$

$-$**Convex Structured Primary Problem**



The negative sign appeared as our definition of $g_\alpha(f, h)$ has a minus sign with respect to the conventional definition of Lagrangian. □

**Lemma E.5** *Given a random variable $Z$ taking values in $[0,1]$ and $\mathbb{E}[Z] \geq \gamma > 0$, we have,*

$$\Pr(Z \geq \frac{\gamma}{3}) \geq \frac{2\gamma}{3}.$$

*Proof.* Let us define $\theta = \Pr(Z \geq \frac{\gamma}{3})$. Now we have,

$$\gamma \leq \mathbb{E}[Z] \leq \theta \times 1 + (1 - \theta) \times \frac{\gamma}{3}.$$

Hence,

$$\theta(1 - \frac{\gamma}{3)} \geq 2\frac{\gamma}{3}.$$

Therefore,

$$\theta \geq \frac{\frac{2\gamma}{3}}{1 - \frac{\gamma}{3}} \geq \frac{2\gamma}{3}. \tag{10}$$

$\square$

**Lemma E.6** *Consider $\delta > 0$, $\frac{1}{10} > \gamma > 0$ and $N$ numbers $z_1, \cdots, z_N$ such that, $\sum_{i=1}^{N} z_i = \gamma$ and $z_i \in [0, \delta]$ for every $1 \leq i \leq N$. Then there exists a subset $S \subseteq [N]$ such that,*

$$\sum_{i \in S} z_i \in [\frac{\gamma^{\frac{1}{2}} \delta^{\frac{1}{4}}}{2}, 2\gamma^{\frac{1}{2}} \delta^{\frac{1}{4}}]. \tag{11}$$

*Proof.* Let us define for every $1 \leq i \leq N$,

$$y_i = \begin{cases} z_i & \text{with probability } \gamma^{\frac{-1}{2}} \delta^{\frac{1}{4}}, \\ 0 & \text{with probability } 1 - \gamma^{\frac{1}{2}} \delta^{\frac{1}{4}}, \end{cases} \tag{12}$$

and assume that $y_i$s are independently generated. By Hoeffding inequality we have,

$$\Pr\left(\left|\sum_{i=1}^{N} y_i - \sum_{i=1}^{N} \mathbb{E}[y_i]\right| \geq \varepsilon\right) \leq 2 \exp\left\{\frac{-2\varepsilon^2}{\sum_{i=1}^{N} z_i^2}\right\}.$$

This results in,

$$\Pr\left(\left|\sum_{i=1}^{N} y_i - \gamma\gamma^{\frac{-1}{2}} \delta^{\frac{1}{4}}\right| \geq \varepsilon\right) \leq 2 \exp\left\{\frac{-2\varepsilon^2}{\delta\gamma}\right\}.$$

Now setting $\varepsilon = \frac{1}{2}\gamma^{\frac{1}{2}} \delta^{\frac{1}{4}}$, we get,

$$\Pr\left(\left|\sum_{i=1}^{N} y_i - \gamma^{\frac{1}{2}} \delta^{\frac{1}{4}}\right| \geq \frac{1}{2}\gamma^{\frac{1}{2}} \delta^{\frac{1}{4}}\right) \leq 2 \exp\left\{\frac{-1}{2\delta^{\frac{1}{2}}}\right\} < 1.$$

This means there exists a deterministic realization that satisfies 11. $\square$

## F    Proofs of Section 3

Here we provide a version of Theorem 3.5, which does not need the assumption 1 . Before that, let us remind that each element $f \in \mathcal{F}$ can be represented by a $\boldsymbol{\beta} \in \mathbb{R}^d$, where we use the notation $f_{\boldsymbol{\beta}}(x) = \langle \boldsymbol{\beta}, \Phi(x) \rangle$ (look at section 2.1 for more details).

**Theorem F.1** *Let us assume $\mathcal{D}_{S|X}$ and $\mathcal{D}_X$ are continuous. Let $\mathcal{F}$ be a bounded finite-dimensional affine class of covariate shifts and $\mathcal{H}$ be the class of all measurable functions. Let $f^*$ denote the optimal solution to the outer minimization and $\mathrm{OPT}$ denotes the optimal value of the Relaxed Minimax Problem 3.2. Finally, let $L^*$ denotes the optimal value for the Relaxed Primary Problem 3.3. For every $\varepsilon > 0$, there exists $h^* \in \mathcal{H}$ such that,*

*(i)* $|g_\alpha(f^*, h^*) - \text{OPT}| \leq \varepsilon$.

*(ii) For every $f_{\boldsymbol{\beta}} \in \mathcal{F}$ we have,*

$$\left| \mathbb{E}\left[ f_{\boldsymbol{\beta}}(X_{n+1}) \Big\{ \mathbf{1}[Y_{n+1} \in C_{h^*}^S(X_{n+1})] - (1-\alpha) \Big\} \right] \right| \leq \varepsilon ||\boldsymbol{\beta}||_1.$$

*(iii)* $\mathbb{E}\,\text{len}(C_{h^*}^S(X)) \leq L^* + \varepsilon$.

Put it simply, Theorem F.1 says fixing any $\varepsilon > 0$, there is an $\varepsilon$-close optimal solution of the Relaxed Minimax Problem (statement (i)) such that it is $\varepsilon$-close to the feasible solutions of the Relaxed Primary Problem which have perfect conditional coverage (statement (ii)), and it achieves an at most $\varepsilon$-larger prediction set length compare to the smallest possible, which is the solution of Relaxed Minimax Problem (statement (iii)).

**Proof of Theorem F.1:** Let us define $\mathcal{H}^\infty$ as the class of all measurable functions from $\mathcal{X}$ to $\mathbb{R}$. Now let us restate the Structured Minimax Problem for $\mathcal{H}^\infty$.

---
**Structured Minimax Problem:**
$$\underset{f \in \mathcal{F}}{\text{Minimize}} \ \underset{h \in \mathcal{H}^\infty}{\text{Maximize}} \ g_\alpha(f, h).$$
---

We can now rewrite this problem in the equivalent form of the following, where we change the domain of the maximization from $\mathcal{H}_\infty$ to corresponding set functions. Let $\mathcal{C}_{\mathcal{H}^\infty}$ be the set of all function from $C(x, y) : \mathcal{X} \times \mathcal{Y} \to \mathbb{R}$ such that there exists $h \in \mathcal{H}^\infty$ such that $C(x, y) = 1[S(x, y) \leq h(x)]$.

---
**Structured Minimax Problem:**
$$\underset{f \in \mathcal{F}}{\text{Minimize}} \ \underset{C \in \mathcal{C}_{\mathcal{H}^\infty}}{\text{Maximize}} \ g_\alpha(f, C).$$
---

We now proceed by defining a convexified version of this problem. Let us define

$$\mathcal{C}_{\mathcal{H}^\infty}^{\text{con}} = \{\sum_{i=1}^{T} a_i C_i(x, y) \mid \sum_{i=1}^{T} a_i = 1, a_i \geq 0 (\forall 1 \leq i \leq T), C_i \in \mathcal{C}_{\mathcal{H}^\infty} (\forall 1 \leq i \leq T), T \in \boldsymbol{N}\}.$$

Now we can define the following problem.

---
**Convex Structured Minimax Problem:**
$$\underset{f \in \mathcal{F}}{\text{Minimize}} \ \underset{C \in \mathcal{C}_{\mathcal{H}^\infty}^{\text{con}}}{\text{Maximize}} \ g_\alpha(f, C).$$
---

Now applying lemma E.4 we have,



**Structured Minimax Problem**

$\equiv$

**Convex Structured Minimax Problem**

$\equiv$

$-$**Convex Structured Primary Problem**



The negative sign appears because our definition of $\boldsymbol{g}_\alpha(f, h)$ has a minus sign with respect to the conventional definition of Lagrangian. Let us call the optimal value for all these three problems OPT (here pay attention that based off of our definition OPT is always a negative number, hence when we are addressing the optimal length the term $-$OPT shows up). With some abuse of notation, Let us assume $\{C_i\}_{i=1}^{\infty}$, where $C_i \in \mathcal{C}_{\mathcal{H}^\infty}^{\text{con}}$ is a feasible sequence of Convex Structured Primary Problem which achieves the optimal value in the limit, i.e, $\lim_{i \to \infty} g_\alpha(f, C_i) = -$OPT. This means, for any $\varepsilon \geq 0$, there is an index $i_\varepsilon$ such that $|\mathbb{E}\,\text{len}(C_{i_\varepsilon}) + \text{OPT}| \leq \varepsilon$. Let us fix the value of $\varepsilon$ for now, we will determine its value later. By definition, there should be $\{\tilde{C}_i\}_{i=1}^{t}$ and corresponding $\{h_i\}_{i=1}^{t}$ such that $\tilde{C}_i \in \mathcal{C}_{\mathcal{H}^\infty}$, $h_i \in \mathcal{H}^\infty$, $\tilde{C}_i = \mathbf{1}[S(x, y) \leq h_i(x)]$, and we have $C_{i_\varepsilon} = \sum_{i=1}^{t} a_i \tilde{C}_i(x, y)$ and $\sum_{i=1}^{t} a_i = 1, a_i \geq 0 \ (\forall 1 \leq i \leq t)$.

Now Let us define $L_i = \mathbb{E}\operatorname{len}(\tilde{C}_i)$ for every $1 \le i \le t$ (all of these expectations are finite cause $|\mathbb{E}\operatorname{len}(C_{i_\varepsilon}) + \operatorname{OPT}| \le \varepsilon$). Now if we look at the truncations of these expectations, by Dominated Convergence Theorem we have, $\mathbb{E}[\operatorname{len}(\tilde{C}_i)\mathbf{1}[\operatorname{len}(\tilde{C}_i) > k]] \xrightarrow[k \to \infty]{} 0$.

This means, for any $\varepsilon \ge 0$, there exists a real valued number $\gamma_1$ such that we can define the set $\Gamma_{\gamma_1} \subset \mathcal{X}$ so that (i) for every $1 \le i \le t$ and for every $x \in \Gamma_{\gamma_1}$, $\operatorname{len}(\tilde{C}_i(x)) \le \gamma_1$, and (ii) $\mathbb{E}[\operatorname{len}(\tilde{C}_{i_\varepsilon})\mathbf{1}[x \notin \Gamma_{\gamma_1}]] \le \varepsilon$.

Now remind that $\mathcal{X} = \mathbb{R}^p$. Since we know $\mathcal{D}_X$ is continuous then by Dominated Convergence Theorem there exists large enough real value $\gamma_2$ such that $\Pr(X \notin \mathrm{B}(0, \gamma_2)) \le \varepsilon$ where $\mathrm{B}(0, \gamma_2)$ is the ball with radius $\gamma_2$ and $\varepsilon$ is a positive real number that we will determine later.

Similarly we can again look at $\tilde{L}_{ik} = L_i\mathbf{1}[||X||_2 \ge k]$. By Dominated Convergence Theorem there is a $\gamma_3$ such that for every $1 \le i \le t$ we have $\mathbb{E}[L_i\mathbf{1}[||X||_2 \ge \gamma_3]] \le \varepsilon$.

Let $R = \max\{\gamma_1, \gamma_2, \gamma_3, 1\}$ and $A = \mathrm{B}(0, R) \cap L_R$.

Now the rest of the proof proceed as follows. We will construct $h^* \in \mathcal{H}^\infty$, and as a result the corresponding set $C^*(x, y) = \mathbf{1}[S(x, y) \le h^*(x)]$, in a way that coverage properties and the length of $C^*$ is close to the ones for $C_{i_\varepsilon}$. Let us proceed with the construction of $h^*$.

**case 1:** $x \notin A$. For this case we define $h^*(x) = 0$.

**case 2:** $x \in A$. This case is more involved. Let us fix $\varepsilon_1 \ge 0$. Let $\{\mathrm{B}(b_i, \varepsilon_1)\}_{i=1}^N$ be a minimal $\varepsilon_1$ covering for $A$. We can then further prune this covering balls to $\{W_i\}_{i=1}^N$, where (i) $W_i \cap W_j = \emptyset$ for every $i \ne j$, (ii) $\bigcup_{i=1}^N W_i = A$, and (iii) $W_i \subseteq \mathrm{B}(b_i, \varepsilon_1)$ for every $i$. Now we use the following randomized method to prove a desirable construction for $h^*$ exists. Let $\{Z_i\}_{i=1}^\infty$ be a collection of uniform iid discrete random variables where they take values between $1$ and $t$. Then we define the random function $h_{rand}(x)$ inside $A$ in the following way. By construction, for each $x \in A$ there is a unique $W_i$ that includes $x$. We set $h_{rand}(x) = h_{Z_i}(x)$. We denote the corresponding set function to $h_{rand}$ by $C_{rand}$. Now Let us remind that $F$ is a finite dimensional affine class of functions. In particular, let $\Phi(x) = [\phi_1(x), \cdots, \phi_d(x)]$ be a *predefined* function (a.k.a. the finite-dimensional basis). The class $\mathcal{F}$ is defined as $\mathcal{F} = \{\langle \boldsymbol{\beta}, \Phi(x)\rangle | \boldsymbol{\beta} \in \mathbb{R}^d\}$. Now for each $k \in [1, \cdots, d]$ we can do the following calculations.

$$
\left| \mathop{\mathbb{E}}_{X,Y,h_{rand}} \left[ \sum_{i=1}^N \phi_k(X)(C_{rand}(X, Y) - (1 - \alpha))\mathbf{1}[X \in W_i] \right] \right|
$$
$$
= \left| \mathop{\mathbb{E}}_{X,Y,h_{rand}} \left[ \phi_k(X)(C_{rand}(X, Y) - (1 - \alpha))\mathbf{1}[X \in A] \right] \right|
$$
$$
= \left| \sum_{i=1}^t a_i \mathop{\mathbb{E}}_{X,Y} \left[ \phi_k(X)(\tilde{C}_i(x, y) - (1 - \alpha))\mathbf{1}[X \in A] \right] \right|
$$
$$
= \left| \mathop{\mathbb{E}}_{X,Y} \left[ \phi_k(X)(C_{i_\varepsilon}(X, Y) - (1 - \alpha))\mathbf{1}[X \in A] \right] \right|
$$
$$
= \left| 0 - \mathop{\mathbb{E}}_{X,Y} \left[ \phi_k(X)(C_{i_\varepsilon}(X, Y) - (1 - \alpha))\mathbf{1}[X \notin A] \right] \right|
$$
$$
\le \mathop{\mathbb{E}}_{X,Y} \left[ \left| \phi_k(X) \right| \left| (C_{i_\varepsilon}(X, Y) - (1 - \alpha)) \right| \left| \mathbf{1}[X \notin A] \right| \right]
$$
$$
\le B\,\mathbb{E}\,\mathbf{1}[X \notin A]
$$
$$
\le B(\mathbb{E}\,\mathrm{B}(0, R) + \mathbb{E}\,L_R)
$$
$$
\le 2B\varepsilon,
$$

where $B$ is the upper bound for $\phi_k$. We now define the random variable

$$
E_i = \mathop{\mathbb{E}}_{X,Y}[\phi_k(X)(C_{rand}(X, Y) - (1 - \alpha))\mathbf{1}[X \in W_i]]
$$

for every $1 \leq i \leq N$. The computation above indicates a bound on the expectation of the sum of these variables. By the construction of $C_{rand}$ we know that they are independent. Furthermore, each random variable $E_i$ is bounded by the quantity $2BM\text{vol}(\text{B}(0, \varepsilon_1))$, where $B$ is the upper bound for $\phi_k$, $M = \max_{x \in \text{B}(0,R)} p(x)$ (which is finite as $\text{B}(0, R)$ is a compact set and we assumed $\mathcal{D}_{\mathcal{X}}$ is continuous), and $\text{B}(0, \varepsilon_1)$ appears as a result of $W_i \subseteq \text{B}(b_i, \varepsilon_1)$. Hence, we can apply Hoeffding inequality for general bounded random variables and derive for every $\varepsilon \geq 0$,

$$\Pr\left(\left|\sum_{i=1}^{N} E_i - \mathbb{E}[\sum_{i=1}^{N} E_i]\right| < \varepsilon\right) \geq 1 - 2\exp\left\{\frac{-2\varepsilon^2}{4NB^2M^2\text{vol}(\text{B}(0, \varepsilon_1))^2}\right\}.$$

This results in,

$$\Pr\left(\left|\underset{X,Y}{\mathbb{E}}\left[\phi_k(X)(C_{rand}(X,Y)-(1-\alpha))\mathbf{1}[X \in A]\right]\right| < \varepsilon + 2B\varepsilon\right)$$
$$\geq 1 - 2\exp\left\{\frac{-2\varepsilon^2}{4NB^2M^2\text{vol}(\text{B}(0, \varepsilon_1))^2}\right\}.$$

Furthermore,

$$\left|\underset{X,Y}{\mathbb{E}}\left[\phi_k(X)(C_{rand}(X,Y)-(1-\alpha))\mathbf{1}[X \notin A]\right]\right|$$
$$= \underset{X,Y}{\mathbb{E}}\left[\left|\phi_k(X)\right|\left|(C_{rand}(X,Y)-(1-\alpha))\right|\mathbf{1}[X \notin A]\right]$$
$$\leq B\,\mathbb{E}\,\mathbf{1}[X \notin A]$$
$$\leq B(\mathbb{E}\,\text{B}(0, R) + \mathbb{E}\,L_R)$$
$$\leq 2B\varepsilon.$$

Therefore,

$$\Pr\left(\left|\underset{X,Y}{\mathbb{E}}\left[\phi_k(X)(C_{rand}(X,Y)-(1-\alpha))\right]\right| < \varepsilon + 4B\varepsilon\right)$$
$$\geq 1 - 2\exp\left\{\frac{-2\varepsilon^2}{4NB^2M^2\text{vol}(\text{B}(0, \varepsilon_1))^2}\right\}.$$

Now since we have this argument for each $1 \leq k \leq d$, by a union bound we have,

$$\Pr\left(\forall k : 1 \leq k \leq d : \left|\underset{X,Y}{\mathbb{E}}\left[\phi_k(X)(C_{rand}(X,Y)-(1-\alpha))\right]\right| < \varepsilon + 4B\varepsilon\right) \tag{13}$$
$$\geq 1 - 2d\exp\left\{\frac{-2\varepsilon^2}{4NB^2M^2\text{vol}(\text{B}(0, \varepsilon_1))^2}\right\}.$$

Now one can argue a similar inequality should hold for length too. This time we can define $Q_i = \underset{X}{\mathbb{E}}\left[\int_{\mathcal{Y}} C_{rand}(X, y)dy\mathbf{1}[X \in W_i]\right]$. Now these variables are also independent and as a result

of the construction of the covering set they are bounded by $RM\mathrm{vol}(\mathrm{B}(0,\varepsilon_1))$. We also have,

$$\left|\underset{X,h_{rand}}{\mathbb{E}}\sum_{i=1}^{N}Q_i\mathbf{1}[X\in W_i]+\mathrm{OPT}\right|=\left|\underset{X,h_{rand}}{\mathbb{E}}\int_{\mathcal{Y}}C_{rand}(X,y)dy\mathbf{1}[X\in A]+\mathrm{OPT}\right|$$

$$=\left|\sum_{i=1}^{t}a_i\mathbb{E}\int_{X}\int_{\mathcal{Y}}C_i(X,y)dy\mathbf{1}[X\in A]+\mathrm{OPT}\right|$$

$$=\left|\mathbb{E}\int_{X}\int_{\mathcal{Y}}C_{i_\varepsilon}(X,y)dy\mathbf{1}[X\in A]-\mathrm{OPT}\right|$$

$$=\left|\mathbb{E}\int_{X}\int_{\mathcal{Y}}C_{i_\varepsilon}(X,y)dy+\mathrm{OPT}\right.$$

$$\left.-\mathbb{E}\int_{X}\int_{\mathcal{Y}}C_{i_\varepsilon}(X,y)dy\mathbf{1}[X\notin A]\right|$$

$$\leq\varepsilon+\mathbb{E}\int_{X}\int_{\mathcal{Y}}C_{i_\varepsilon}(X,y)dy\mathbf{1}[X\notin A]$$

$$\leq\varepsilon+\mathbb{E}\int_{X}\int_{\mathcal{Y}}C_{i_\varepsilon}(X,y)dy\mathbf{1}[X\notin\mathrm{B}(0,R)]$$

$$+\mathbb{E}\int_{X}\int_{\mathcal{Y}}C_{i_\varepsilon}(X,y)dy\mathbf{1}[X\notin L_R)]$$

$$\leq3\varepsilon.$$

Now again by applying Hoeffding inequality for general bounded random variables we have for every $\varepsilon\geq0$,

$$\Pr(\left|\sum_{i=1}^{N}Q_i-\mathbb{E}[\sum_{i=1}^{N}Q_i]\right|<\varepsilon)\geq1-2\exp\left\{\frac{-2\varepsilon^2}{NR^2M^2\mathrm{vol}(\mathrm{B}(0,\varepsilon_1))^2}\right\}.$$

This results in,

$$\Pr(\left|\mathbb{E}\int_{X}\int_{\mathcal{Y}}C_{rand}(X,y)dy\mathbf{1}[X\in A]+\mathrm{OPT}\right|<4\varepsilon)\geq1-2\exp\left\{\frac{-2\varepsilon^2}{NR^2M^2\mathrm{vol}(\mathrm{B}(0,\varepsilon_1))^2}\right\}.$$

Furthermore, by construction of $C_{rand}$,

$$\left|\mathbb{E}\int_{X}\int_{\mathcal{Y}}C_{rand}(X,y)dy\mathbf{1}[X\notin A]\right|=0.$$

Therefore,

$$\Pr(\left|\mathbb{E}\int_{X}\int_{\mathcal{Y}}C_{rand}(X,y)dy+\mathrm{OPT}\right|<4\varepsilon)\geq1-2\exp\left\{\frac{-2\varepsilon^2}{NR^2M^2\mathrm{vol}(\mathrm{B}(0,\varepsilon_1))^2}\right\}.\qquad(14)$$

Now a union bound between (13) and (14) leads to,

$$\Pr\left(\forall k:1\leq k\leq d:\left|\underset{X,Y}{\mathbb{E}}\left[\phi_k(X)(C_{rand}(X,Y)-(1-\alpha))\right]\right|<\varepsilon+4B\varepsilon\quad\text{and}\right.$$

$$\left.\left|\mathbb{E}\int_{X}\int_{\mathcal{Y}}C_{rand}(X,y)dy+\mathrm{OPT}\right|<4\varepsilon\right)$$

$$\geq1-2d\exp\left\{\frac{-2\varepsilon^2}{4NB^2M^2\mathrm{vol}(\mathrm{B}(0,\varepsilon_1))^2}\right\}-2\exp\left\{\frac{-2\varepsilon^2}{NR^2M^2\mathrm{vol}(\mathrm{B}(0,\varepsilon_1))^2}\right\}.$$

Recalling, $N=\mathcal{N}(A,\|.\|_2,\varepsilon_1)$, where $\mathcal{N}$ denotes the covering number, we have the following inequality (look at chapter 4 of [66]),

$$N\leq(\frac{3}{\varepsilon_1})^p\frac{\mathrm{vol}(A)}{\mathrm{vol}(\mathrm{B}(0,1))},\quad\mathrm{vol}(\mathrm{B}(0,\varepsilon_1))=\varepsilon_1^p\mathrm{vol}(\mathrm{B}(0,1)).$$

This results in,

$$\Pr\left(\forall k: 1 \leq k \leq d: \left|\underset{X,Y}{\mathbb{E}}\left[\phi_k(X)(C_{rand}(X,Y)-(1-\alpha))\right]\right| < \varepsilon + 4B\varepsilon \quad \text{and}\right.$$

$$\left.\left|\underset{X}{\mathbb{E}}\int_{\mathcal{Y}} C_{rand}(X,y)dy + \text{OPT}\right| < 4\varepsilon\right)$$

$$\geq 1 - 2d\exp\left\{\frac{-2\varepsilon^2}{3^p 4\text{vol}(A)B^2 M^2 \varepsilon_1^p \text{vol}(\text{B}(0,1))}\right\} - 2\exp\left\{\frac{-2\varepsilon^2}{3^p \text{vol}(A)R^2 M^2 \varepsilon_1^p \text{vol}(\text{B}(0,1))}\right\}.$$

Since we have this inequality for every $\varepsilon_1 > 0$, we can pick a small enough $\varepsilon_1$ such that,

$$\Pr\left(\forall k: 1 \leq k \leq d: \left|\underset{X,Y}{\mathbb{E}}\left[\phi_k(X)(C_{rand}(X,Y)-(1-\alpha))\right]\right| < \varepsilon + 4B\varepsilon \quad \text{and}\right.$$

$$\left.\left|\underset{X}{\mathbb{E}}\int_{\mathcal{Y}} C_{rand}(X,y)dy + \text{OPT}\right| < 4\varepsilon\right) \geq \frac{1}{2}.$$

This means there is a realization of $h_{rand}$ and accordingly $C_{rand}$, which we denote them by $h^*$ and $C^*$ such that,

$$\forall k: 1 \leq k \leq d: \left|\underset{X,Y}{\mathbb{E}}\left[\phi_k(X)(C^*(X,Y)-(1-\alpha))\right]\right| < \varepsilon + 4B\varepsilon \quad \text{and}$$

$$\left|\underset{X}{\mathbb{E}}\int_{\mathcal{Y}} C^*(X,y)dy + \text{OPT}\right| < 4\varepsilon.$$

Now since we proved this for any $\varepsilon > 0$, we can put $\varepsilon = \varepsilon' \min\{\frac{1}{4}, \frac{1}{1+4B}\}$. Therefore the following statement holds. For every $\varepsilon' > 0$ there exists $h^* \in \mathcal{H}^\infty$ and its corresponding set function $C^*(x,y) = \mathbf{1}[S(x,y) \leq h^*(x)]$ such that,

$$\forall k: 1 \leq k \leq d: \left|\underset{X,Y}{\mathbb{E}}\left[\phi_k(X)(C^*(X,Y)-(1-\alpha))\right]\right| < \varepsilon' \quad \text{and}$$

$$\left|\underset{X}{\mathbb{E}}\int_{\mathcal{Y}} C^*(X,y)dy + \text{OPT}\right| < \varepsilon'.$$

This immediately proves the statement (ii) of the Theorem 3.5. The statement (iii) also follows by the fact that by weak duality between the Relaxed Minimax Problem and the Relaxed Primary Problem we have $-\text{OPT} \leq L^*$. Finally, let $f^*$ denotes an optimal solution to the outer minimization of the Relaxed Minimax Problem. There exists a $\boldsymbol{\beta}^*$ such that $f^*(x) = \langle \boldsymbol{\beta}^*, \Phi(x) \rangle$. Now we have,

$$\left|g_\alpha(f^*, h^*) - \text{OPT}\right| = \left|\mathbb{E}\left[\langle \boldsymbol{\beta}^*, \Phi(X) \rangle \left\{C^*(X,Y) - (1-\alpha)\right\}\right] - \mathbb{E}\int_{\mathcal{Y}} C^*(X,y)dy\right|$$

$$\leq \left|\mathbb{E}\left[\langle \boldsymbol{\beta}^*, \Phi(X) \rangle \left\{C^*(X,Y) - (1-\alpha)\right\}\right]\right| + \left|\mathbb{E}\int_{\mathcal{Y}} C^*(X,y)dy + \text{OPT}\right|$$

$$\leq \varepsilon' \|\boldsymbol{\beta}^*\|_1 + \varepsilon'.$$

Hence $\varepsilon' \leq \frac{\varepsilon}{1+\|\boldsymbol{\beta}^*\|_1}$ prove the statement (i) of the Theorem 3.5.

**Proof of Theorem 3.5:** Let us define $\mathcal{H}^\infty$ as the class of all measurable functions from $\mathcal{X}$ to $\mathbb{R}$. Now let us restate the Structured Minimax Problem for $\mathcal{H}^\infty$.

---

**Structured Minimax Problem:**

$$\underset{f \in \mathcal{F}}{\text{Minimize}} \ \underset{h \in \mathcal{H}^\infty}{\text{Maximize}} \ g_\alpha(f, h).$$

---

We can now rewrite this problem in the equivalent form of the following, where we change the domain of the maximization from $\mathcal{H}_\infty$ to corresponding set functions. Let $\mathcal{C}_{\mathcal{H}^\infty}$ be the set of all function from $C(x,y) : \mathcal{X} \times \mathcal{Y} \to \mathbb{R}$ such that there exists $h \in \mathcal{H}^\infty$ such that $C(x,y) = 1[S(x,y) \leq h(x)]$.

**Structured Minimax Problem:**
$$\underset{f \in \mathcal{F}}{\text{Minimize}} \ \underset{C \in \mathcal{C}_{\mathcal{H}^\infty}}{\text{Maximize}} \ g_\alpha(f, C).$$

We now proceed by defining a convexified version of this problem. Let us define

$$\mathcal{C}_{\mathcal{H}^\infty}^{\text{con}} = \{\sum_{i=1}^{T} a_i C_i(x, y) \mid \sum_{i=1}^{T} a_i = 1, a_i \geq 0 (\forall 1 \leq i \leq T), C_i \in \mathcal{C}_{\mathcal{H}^\infty} (\forall 1 \leq i \leq T), T \in \boldsymbol{N}\}.$$

Now we can define the following problem.

**Convex Structured Minimax Problem:**
$$\underset{f \in \mathcal{F}}{\text{Minimize}} \ \underset{C \in \mathcal{C}_{\mathcal{H}^\infty}^{\text{con}}}{\text{Maximize}} \ g_\alpha(f, C).$$

Naturally we can also define the Convex Structured Primary Problem.

**Convex Structured Primary Problem:**
$$\underset{C \in \mathcal{C}_{\mathcal{H}^\infty}^{\text{con}}}{\text{Minimize}} \quad \mathbb{E}\left[\text{len}(C(X))\right]$$
$$\text{subject to} \quad \mathbb{E}\left[f(X)\Big\{C(X, Y) - (1 - \alpha)\Big\}\right] = 0, \quad \forall f \in \mathcal{F}$$

Now applying lemma E.4 we have,



**Structured Minimax Problem**

$\equiv$

**Convex Structured Minimax Problem**

$\equiv$

$-$**Convex Structured Primary Problem**



Let us call the optimal value for all these three problems OPT (here pay attention that based off of our definition OPT is always a negative number, hence when we are addressing the optimal length the term $-$OPT shows up). With some abuse of notation, Let us assume $\{C_i\}_{i=1}^\infty$, where $C_i \in \mathcal{C}_{\mathcal{H}^\infty}^{\text{con}}$ is a feasible sequence of Convex Structured Primary Problem which achieves the optimal value in the limit, i.e, $\lim_{i \to \infty} g_\alpha(f, C_i) = -$OPT. This means, for any $\varepsilon \geq 0$, there is an index $i_\varepsilon$ such that $|\mathbb{E}\,\text{len}(C_{i_\varepsilon}) + \text{OPT}| \leq \varepsilon$. Let us fix the value of $\varepsilon$ for now, we will determine its value later. By definition, there should be $\{\tilde{C}_i\}_{i=1}^t$ and corresponding $\{h_i\}_{i=1}^t$ such that $\tilde{C}_i \in \mathcal{C}_{\mathcal{H}^\infty}$, $h_i \in \mathcal{H}^\infty$, $\tilde{C}_i = \mathbf{1}[S(x, y) \leq h_i(x)]$, and we have $C_{i_\varepsilon} = \sum_{i=1}^{t} a_i \tilde{C}_i(x, y)$ and $\sum_{i=1}^{t} a_i = 1, a_i \geq 0 \ (\forall 1 \leq i \leq t)$.

Let us recall that the class $\mathcal{F}$ is defined as $\mathcal{F} = \{\langle \boldsymbol{\beta}, \Phi(x) \rangle | \boldsymbol{\beta} \in \mathbb{R}^d\}$, where $\Phi : \mathcal{X} \to \mathbb{R}^d$ is a *predefined* function (a.k.a. the finite-dimensional basis). Now let $\{\Phi_i\}_{i=1}^\infty$ be all the possible countable values that it takes. In other words, each $\Phi_i$ is a vector in $\mathbb{R}^d$ and there exists $x \in \mathcal{X}$ such that $\Phi(x) = \Phi_i$. Then we have,

$$\int_{\mathcal{X}} \sum_{i=1}^{t} a_i \Phi(x) \left(\int_{\mathcal{Y}} \mathbf{1}[S(x, y) \leq h_i(x)] p(y|x) dy\right) p(x) dx = (1 - \alpha) \begin{pmatrix} 1 \\ 1 \\ \vdots \\ 1 \end{pmatrix}. \tag{15}$$

Here we implicitly assumed that $\Phi(x)$ is properly normalized so that the Right hand side don't need any extra normalization. Let us define,

$$\text{cov}_i(x) = \int_{\mathcal{Y}} \mathbf{1}[S(x, y) \leq h_i(x)] p(y|x) dy,$$

$$l_i(x) = \int_{\mathcal{Y}} \mathbf{1}[S(x,y) \leq h_i(x)]dy.$$

from (15) we have,

$$\int_{\mathcal{X}} \sum_{i=1}^{t} a_i \Phi(x)\mathrm{cov}_i(x)p(x)dx = (1-\alpha)\begin{pmatrix} 1 \\ 1 \\ \vdots \\ 1 \end{pmatrix}.$$

We can then write,

$$\sum_{j=1}^{\infty} \Phi_j \int_{\mathcal{X}_j} \left( \sum_{i=1}^{t} a_i\mathrm{cov}_i(x) \right) p(x)dx = (1-\alpha)\begin{pmatrix} 1 \\ 1 \\ \vdots \\ 1 \end{pmatrix}, \qquad (16)$$

where $\mathcal{X}_j = \{x \in \mathcal{X} \mid \Phi(x) = \Phi_j\}$. Now two points are in order. (i) Without loss of generality, one can assume $p(\mathcal{X}_j) > 0$ for every $j$ as otherwise we can just omit that $\Phi_j$ since it does not contribute to any integral due to continuity assumptions. (ii) $\{\mathcal{X}_j\}_{j=1}^{\infty}$ partitions $\mathcal{X}$, i.e., their union is $\mathcal{X}$ and they are pairwise disjoint. Now the idea behind the rest of the proof is we show for each region $\mathcal{X}_j$, there is a single function $h \in \mathcal{H}$ such that,

$$(i) \int_{\mathcal{X}_j} \left( \sum_{i=1}^{t} a_i\mathrm{cov}_i(x) \right) p(x)dx = \int_{\mathcal{X}_i} \mathrm{cov}_h(x)p(x)dx,$$

$$\text{where } \mathrm{cov}_h(x) = \int_{\mathcal{Y}} \mathbf{1}[S(x,y) \leq h(x)]p(y|x)dy.$$

$$(ii) \int_{\mathcal{X}_j} l_h(x)p(x)dx \leq \int_{\mathcal{X}_j} \left( \sum_{i=1}^{t} a_i l_i(x) \right) p(x)dx + \varepsilon p(\mathcal{X}_j),$$

$$\text{where } l_h(x) = \int_{\mathcal{Y}} \mathbf{1}[S(x,y) \leq h(x)]dy.$$

Hence, from now on we fix an arbitrary $\mathcal{X}_j$ and prove the existence of such $h$. For the ease of notation let us define $\tilde{\mathcal{X}} = \mathcal{X}_j$ and $\tilde{p}(x) = \frac{p(x)}{p(\tilde{\mathcal{X}})}$. This way $\int_{\tilde{\mathcal{X}}} \tilde{p}(x) = 1$ and $\tilde{p}(x)$ is still continuous.

Now we can rewrite our goals with this new notation. With a simple normalization we have,

$$(i) \int_{\tilde{\mathcal{X}}} \left( \sum_{i=1}^{t} a_i\mathrm{cov}_i(x) \right) \tilde{p}(x)dx = \int_{\tilde{\mathcal{X}}} \mathrm{cov}_h(x)\tilde{p}(x)dx,$$

$$\text{where } \mathrm{cov}_h(x) = \int_{\mathcal{Y}} \mathbf{1}[S(x,y) \leq h(x)]p(y|x)dy.$$

$$(ii) \int_{\tilde{\mathcal{X}}} l_h(x)\tilde{p}(x)dx \leq \int_{\tilde{\mathcal{X}}} \left( \sum_{i=1}^{t} a_i l_i(x) \right) \tilde{p}(x)dx + \varepsilon,$$

$$\text{where } l_h(x) = \int_{\mathcal{Y}} \mathbf{1}[S(x,y) \leq h(x)]dy.$$

Now, as a result of Dominated Convergence Theorem, there is a large enough real value $R$ such that for the set of $\tilde{\mathcal{X}}_1 = \{x \in \tilde{\mathcal{X}} \mid ||x||_2 \leq R, \max_{i \in [1,\cdots,t]} l_i(x) \leq R\}$ we have,

$$\sum_{i=1}^{t} \int_{\tilde{\mathcal{X}}} l_i(x)\mathbf{1}[x \notin \tilde{\mathcal{X}}_1]\tilde{p}(x)dx \leq \varepsilon_1, \qquad (17)$$

$$\text{and} \sum_{i=1}^{t} \int_{\tilde{\mathcal{X}}} \mathrm{cov}_i(x)\mathbf{1}[x \notin \tilde{\mathcal{X}}_1]\tilde{p}(x)dx \leq \varepsilon_1, \qquad (18)$$

where the value of $\varepsilon_1 > 0$ will be chosen later in a way that it would be small enough for the proof to work. Among the functions $\mathrm{cov}_1, \cdots, \mathrm{cov}_t$ there should be at least two of them, without loss of generality $\mathrm{cov}_1$ and $\mathrm{cov}_2$ so that $\int_{\tilde{\mathcal{X}}} \mathrm{cov}_1(x)\tilde{p}(x)dx \geq \int_{\tilde{\mathcal{X}}} \mathrm{cov}_2(x)\tilde{p}(x)dx + \gamma$ where $\gamma > 0$. Otherwise we have alreasy achieved our goal by picking the $h_i$ with the smallest average length. Now assume $\varepsilon_1 < \frac{\gamma}{4}$ (we will make sure to pick $\varepsilon_1$ in a way that it satisfies this condition). Hence we have,

$$\int_{\tilde{\mathcal{X}}_1} \mathrm{cov}_1(x)\tilde{p}(x)dx \geq \int_{\tilde{\mathcal{X}}_1} \mathrm{cov}_2(x)\tilde{p}(x)dx + \frac{\gamma}{4}. \tag{19}$$

Now applying Lemma E.5 there exists $\tilde{\mathcal{X}}_2 \subseteq \tilde{\mathcal{X}}_1$ such that $\tilde{p}(\tilde{\mathcal{X}}_2) \geq \frac{\gamma}{6}$ and for every $x \in \tilde{\mathcal{X}}_2$ we have $\mathrm{cov}_1(x) \geq \mathrm{cov}_2(x) + \frac{\gamma}{12}$. (Here one have to use Lemma E.5 using $Z = (\mathrm{cov}_1(x) - \mathrm{cov}_2(x))\,\mathbf{1}[\mathrm{cov}_1(x) \geq \mathrm{cov}_2(x)]\mathbf{1}[x \in \tilde{\mathcal{X}}_1])$

Let us consider the following three sets that partition the space $\tilde{\mathcal{X}}$,

$$\begin{aligned} A &= \tilde{\mathcal{X}}_2 \\ B &= \tilde{\mathcal{X}}_1 \backslash \tilde{\mathcal{X}}_2 \\ C &= \tilde{\mathcal{X}} \backslash \tilde{\mathcal{X}}_1 \end{aligned} \tag{20}$$

Note that these three sets are pair-wise disjoint and cover the space $\tilde{\mathcal{X}}$. LEt us first consider the set $C$. we would like to consider a function $h_C$ such that,

$$\int_C \left( \sum_{i=1}^t a_i \mathrm{cov}_i(x) \right) \tilde{p}(x)dx = \int_C \mathrm{cov}_{h_C}(x)\tilde{p}(x)dx,$$

$$\text{where } \mathrm{cov}_{h_C}(x) = \int_{\mathcal{Y}} \mathbf{1}[S(x,y) \leq h_C(x)]p(y|x)dy.$$

To do so, without loss of generality, we assume,

$$\int_C \mathrm{cov}_1(x)\tilde{p}(x)dx \geq \int_C \mathrm{cov}_2(x)\tilde{p}(x)dx \geq \cdots \geq \int_C \mathrm{cov}_t(x)\tilde{p}(x)dx.$$

Hence,

$$\int_C \mathrm{cov}_1(x)\tilde{p}(x)dx \geq \int_C \left( \sum_{i=1}^t a_i \mathrm{cov}_i(x) \right) \tilde{p}(x)dx \geq \int_C \mathrm{cov}_t(x)\tilde{p}(x)dx.$$

For $r \in [0, \infty)$ let,

$$f(r) = \int_{C \cap \mathbf{B}(0,r)} \mathrm{cov}_t(x)\tilde{p}(x)dx + \int_{C \backslash \mathbf{B}(0,r)} \mathrm{cov}_1(x)\tilde{p}(x)dx.$$

Note that $f(0) = \int_C \mathrm{cov}_1(x)\tilde{p}(x)dx$ and $f(\infty) = \int_C \mathrm{cov}_t(x)\tilde{p}(x)dx$. Also, as a result of continuity assumptions, $f(r)$ is a continuous function, therefore we can apply Intermediate Value Theorem. As a result, there should be $r_0 < \infty$ such that,

$$f(r_0) = \int_C \left( \sum_{i=1}^t a_i \mathrm{cov}_i(x) \right) \tilde{p}(x)dx.$$

We then naturally pick $h_c$ to be,

$$h_C(x) = \begin{cases} h_t(x) & \text{if } x \in C \cap \mathbf{B}(0,r), \\ h_1(x) & \text{if } x \in C \backslash \mathbf{B}(0,r). \end{cases} \tag{21}$$

Let us now consider the sets $A$ and $B$. Note that by construction they both are a subset of $\mathbf{B}(0,R)$. Now let us consider a $\delta$-net for $A$ and a separate $\delta$-net for $B$. Similar to our arguments in the proof of Theorem F.1, we can construct these $\delta$-nets in a way that they partition $A$ and $B$. That is to say we have, $A = \bigcup_{j=1}^{N_A} A_j$ and $B = \bigcup_{j=1}^{N_B} B_j$ such that withing each $\delta$-net the pair wise intersections are empty. Now, for each $A_j$ (and $B_j$) we choose independently one of $\{h_i\}_{i=1}^t$ with probabilities $\{a_i\}_{i=1}^t$ at random.

Let $J(A, j)$ (similarly $J(B, j)$) be the index of the function $h_i$ assigned to $A_j$ (similarly $B_j$). Then we have,

**Event 1:**

$$\left| \sum_{j=1}^{N_A} \int_{A_j} \mathrm{cov}_{h_{J(A,j)}} \tilde{p}(x)dx + \sum_{j=1}^{N_B} \int_{B_j} \mathrm{cov}_{h_{J(B,j)}} \tilde{p}(x)dx - \int_{A \cap B} \sum_{i=1}^{t} a_i \mathrm{cov}_i(x)\tilde{p}(x)dx \right|$$
$$\leq \sqrt{\delta}.$$

**Event 2:**

$$\sum_{j:J(A,j)=1} \tilde{p}(A_j) \geq \frac{a_1}{2}\tilde{p}(A).$$

**Event 3:**

$$\sum_{j:J(A,j)=2} \tilde{p}(A_j) \geq \frac{a_2}{2}\tilde{p}(A).$$

Now we can repeat the probabilistic argument of proof of Theorem F.1 here. The key insight is all of the three above-mentioned events are high probability events, in a way that by letting $\delta$ (the precision of the covering net) to be sufficiently small then the probability of these events happening approaches to 1. Hence, there is a deterministic realization that make all these events happen. Now on, we fix that deterministic realization. Particularly, this means we have a realization that satisfies all the events for arbitrary small $\delta$. Now the idea is to make a small change in the configuration that comes from this realization that make the approximate coverage of event 1 to be an exact coverage. With a little abuse of notation, let $J(A, j)$ (similarly $J(B, j)$) be the assignments of that deterministic realization. Without loss of generality, we can assume,

$$\left| \sum_{j=1}^{N_A} \int_{A_j} \mathrm{cov}_{h_{J(A,j)}} \tilde{p}(x)dx + \sum_{j=1}^{N_B} \int_{B_j} \mathrm{cov}_{h_{J(B,j)}} \tilde{p}(x)dx \right| = \int_{A \cap B} \sum_{i=1}^{t} a_i \mathrm{cov}_i(x)\tilde{p}(x)dx - \varepsilon_2, \tag{22}$$

where $\varepsilon_2 \geq 0$ and $\varepsilon_2 \leq$ constant $\times \sqrt{\delta}$ (the case of $\varepsilon_2 \leq 0$ can be similarly handled). That is to say, our deterministic assignment $J(A, j)$ and $J(B, j)$ led to a small under-coverage of $\varepsilon_2$. The idea now is to "engineer" the assignments of some of the $A_j$s in a way that we make the coverage exact while not changing the average length of the current assignment significantly. Remind the definition the event 3 we defined above. Recall that $a_2$ and $\tilde{p}(A)$ are strictly positive numbers. Hence, as a result of (19) and (20) we have, $\tilde{p}(A_2) \geq \frac{\gamma}{12}$. Now we can argue through Lemma E.6. Consider the set $Q = \{j \in [N_a] \mid J(A, j) = 2\}$. We know from event 3 that $\tilde{p}(Q) \geq \frac{a_2}{2}\tilde{p}(A) \geq \frac{a_2\gamma}{24}$. Now let for every $j \in Q : Z_j = \tilde{p}(A_j)$. We know that $Z_j \leq \delta$ and $\gamma' \equiv \sum_{j \in Q} Z_j \geq \frac{a_2\gamma}{24}$. Applying Lemma E.6 there exists a $Q' \subseteq Q$ such that $\sum_{j \in Q'} \tilde{p}(A_j) \in [\frac{\gamma'^{\frac{1}{2}}\delta^{\frac{1}{4}}}{2}, 2\gamma'^{\frac{1}{2}}\delta^{\frac{1}{4}}]$. Recall that since $Q' \subseteq Q$, then for any $j \in Q'$ we have $J(A, j) = 2$. now if we reassign all the $A_j$'s, $j \in Q'$, to $h_1$ (i,e changing $J(A, j)$ to 1 for every $j \in Q'$) then the amount of added coverage would be, $\sum_{j \in S'} \tilde{p}(A_j) \times \frac{\gamma}{12}$, where $\frac{\gamma}{12}$ appears following (19) and the fact that $A_i \in \tilde{\mathcal{X}}_2$. Hence, the amount of added coverage would be bounded by,

$$\sum_{j \in S'} \tilde{p}(A_j) \times \frac{\gamma}{12} \geq \frac{\gamma\gamma'^{\frac{1}{2}}\delta^{\frac{1}{4}}}{2} \geq \sqrt{\frac{a_2}{24}}\frac{\gamma^{\frac{3}{2}}\delta^{\frac{1}{4}}}{2}. \tag{23}$$

Recall that we can pick $\delta$ as small as we want. Hence, we can pick $\delta$ small enough that we have,

$$\sqrt{\frac{a_2}{24}}\frac{\gamma^{\frac{3}{2}}\delta^{\frac{1}{4}}}{2} \geq \sqrt{\delta} \geq \varepsilon_2, \tag{24}$$

where the last inequality follows from the definition of $\varepsilon_2$ in (22). This can be done by letting $\delta \leq \left( \sqrt{\frac{a_2}{24}}\frac{\gamma^{\frac{3}{4}}}{2} \right)^4$. Now by noting (23) and (24) it should be clear that by reassigning all any $A_j$,

$j \in Q'$, the total coverage added will be larger than $\varepsilon_2$ (hence in total we will over-cover). Now we can apply Intermediate Value Theorem once again. Let us define for $r \in [0, \infty]$,

$$f(r) = \sum_{j \in Q'} \int_{A_j \cap \mathbf{B}(0,r)} \text{cov}_1(x) \tilde{p}(x) dx + \int_{A_j \setminus \mathbf{B}(0,r)} \text{cov}_2(x) \tilde{p}(x) dx.$$

Here note that again as a consequence of continuity assumptions $f$ is a continuous function. Also, $f(R) - f(0) \geq \varepsilon_2$. Hence there exists $r_0 \in [0, R]$ such that, $f(r_0) - f(0) = \varepsilon_2$. That is to say we can make the coverage exactly valid. Now let us analyze the deviation in length. Not that for $x \in A \cup B$ we have length is bounded by $R$. This means by reassigning the elements in the set $A_j$, $j \in Q'$, the change in length is at most $R \times \sum_{j \in Q'} \tilde{p}(A_j) \leq 2R\gamma'^{\frac{1}{2}}\delta^{\frac{1}{4}}$, where the last inequaity comes from the fact that $\sum_{j \in Q'} \tilde{p}(A_j) \in [\frac{\gamma'^{\frac{1}{2}}\delta^{\frac{1}{4}}}{2}, 2\gamma'^{\frac{1}{2}}\delta^{\frac{1}{4}}]$. Here again by choosing $\delta$ small enough, particularly $\delta \leq \left(\frac{\varepsilon}{2R\gamma'^{\frac{1}{2}}}\right)^4$, we can ensure the change in length is smaller than $\varepsilon$, hence we achieved our goal.

**Proof of Proposition 3.7:** Let $h^* \in \mathcal{H}$ be an optimal solution of the Relaxed Minimax Problem and the Relaxed Primary Problem, when the optimization is over all the measurable functions from $\mathcal{X}$ to $\mathbb{R}$. Let us denote the corresponding optimal solution to the outer minimization of the Relaxed Minimax Problem by $f^*$. Recall that $\mathcal{F}$ is a finite dimensional affine class of functions. That is to say there exists a vector $\boldsymbol{\beta^*}$ such that $f^*(x) = \langle \boldsymbol{\beta^*}, \Phi(x) \rangle \equiv f_{\boldsymbol{\beta^*}}(x)$. Since $h^*$ is a solution of the Relaxed Primary Problem so it should be conditionally valid with respect to class $\mathcal{F}$. Therefore, as a result of Lemma 3.4, we should have,

$$\nabla_{\boldsymbol{\beta}} \, g_\alpha(f_{\boldsymbol{\beta}}, h^*)\Big|_{\boldsymbol{\beta^*}} = \vec{0}. \tag{25}$$

Now define $\text{obj}(\boldsymbol{\beta}) = \max_{\theta \in \Theta} g_\alpha(f_{\boldsymbol{\beta}}, h_\theta)$. Since $\Theta$ is a compact set and $g_\alpha(f_{\boldsymbol{\beta}}, h_\theta)$ is convex (in fact linear) in $\boldsymbol{\beta}$ so the Danskin's theorem applies to $\text{obj}(\boldsymbol{\beta})$. Therefore, as a result of (25),

$$\vec{0} \in \partial_{\boldsymbol{\beta}} \, \text{obj}(\boldsymbol{\beta})\Big|_{\boldsymbol{\beta^*}}. \tag{26}$$

Now again as a result of Danskin's theorem, $\text{obj}(\boldsymbol{\beta})$ is convex in $\boldsymbol{\beta}$. Therefore, as a consequence of (26), $\boldsymbol{\beta^*}$ is a minimizer of $\text{obj}(\boldsymbol{\beta})$. That is to say, $(f^*, h^*)$ is an optimal solution to the Relaxed Minimax Problem. On the other hand, $h^*$ is also a solution to the Relaxed Primary Problem (as it is a solution to the Relaxed Primary Problem when solved over all the measurable functions). Putting everything together, $h^*$ is a joint optimal solution to both the Relaxed Minimax Problem and the Relaxed Primary Problem, hence we have strong duality.

**Proof of Proposition 3.3:** Let us start by recalling the definitions,

$$g_\alpha(f, C) = \mathbb{E}\left[ f(X)\left\{ \mathbf{1}[Y \in C(X)] - (1 - \alpha) \right\} \right] - \mathbb{E} \int_{\mathcal{Y}} \mathbf{1}[y \in C(X)] dy$$

where

$$C_f(x) = \{y \in \mathcal{Y} \mid f(x)p(y|x) \geq 1\}.$$

Now, fixing $f \in \mathcal{F}$, all we have to show is that $g_\alpha(f, C_f(x)) - g_\alpha(f, C(x)) \geq 0$ for every $C(x) : \mathcal{X} \to 2^{\mathcal{Y}}$.

**Claim 2** $g_\alpha(f, C_f(x)) - g_\alpha(f, C(x)) \geq 0$ *for every* $C(x) : \mathcal{X} \to 2^{\mathcal{Y}}$

*proof:* We have,

$$g_\alpha(f, C_f(x)) - g_\alpha(f, C(x)) \overset{(a)}{=} \mathbb{E}_X \int_{\mathcal{Y}} (f(X)p(y|X) - 1)\left\{ (\mathbf{1}[y \in C_f(X)] - \mathbf{1}[y \in C(X)]) \right\} dy$$

$$\overset{(b)}{=} \mathbb{E}_X \int_{\mathcal{Y}} (f(X)p(y|X) - 1)\left\{ \mathbf{1}[y \in C_f(X) \backslash C(X)] - \mathbf{1}[y \in C(X) \backslash C_f(X)] \right\} dy$$

$$\overset{(c)}{=} \mathbb{E}_X \int_{\mathcal{Y}} \left| (f(X)p(y|X) - 1) \right| \left\{ \mathbf{1}[y \in C_f(X) \Delta C(X)] \right\} dy$$

$$\geq 0,$$

where, (a) follows from the definitions, (b) follows from the definition of set difference operation ($\backslash$) where, $A \backslash B = \{x \mid x \in A \text{ and } x \notin B\}$, and (c) comes from the definition of $C_f(x)$. The proof is complete now. One can define,

$$\mathbb{E}_X[\mathrm{d}(C_f, C)] \equiv \mathbb{E}_X \int_{\mathcal{Y}} \left| (f(X)p(y|X) - 1) \right| \left\{ \mathbf{1}[y \in C_f(X) \Delta C(X)] \right\} dy,$$

and rewrite the above calculation as,

$$g_\alpha(f, C(x)) = g_\alpha(f, C_f(x)) - \mathbb{E}_X[\mathrm{d}(C_f, C)].$$

This reformulation is very intuitive. at a high level, it says maximizing $g_\alpha(f, C(x))$ over $C$, is equivalent to minimizing a distance between $C$ and $C_f$.

**Proof of Proposition 3.1:** Let us start by restating the proposition.

**Proposition F.2** *The Primary Problem and the Minimax Problem are equivalent. Let $(f^*, C^*(x))$ be the optimal solution of Minimax Problem. Then, $C^*$ is also the optimal solution of the PP. Furthermore, $C^*$ has the following form:*

$$C^*(x) = \{y \in \mathcal{Y} \mid f^*(x)p(y|x) \geq 1\} \tag{27}$$

Let us also recall the definition of Primary Problem,

---

**Primary Problem (PP):**

$$\underset{C(x)}{\text{Minimize}} \quad \mathbb{E}\left[\text{len}(C(X))\right]$$

$$\text{subject to} \quad \mathbb{E}\left[f(X)\left\{\mathbf{1}[Y \in C(X)] - (1 - \alpha)\right\}\right] = 0, \quad \forall f \in \mathcal{F}$$

---

Recall that in this paper we assume that $\mathcal{F} = \{\langle \beta, \Phi(x) \rangle \mid \beta \in \mathbb{R}^d\}$ is a finite dimensional affine class of functions (see section 2.1).

Assuming $\Phi = [\phi_1, \phi_2, \cdots, \phi_d]$. The Primary Problem can be equivalently written in terms of $d$ linear constraints on the prediction sets $C(x)$.

---

**Primary Problem- finite constraints:**

$$\underset{C(x)}{\text{Minimize}} \quad \mathbb{E}\left[\text{len}(C(X))\right]$$

$$\text{subject to} \quad \mathbb{E}\left[\phi_i(X)\left\{\mathbf{1}[Y \in C(X)] - (1 - \alpha)\right\}\right] = 0, \quad \forall i \in [1, d].$$

---

We now take a closer look at the main optimization variable, i.e. the prediction sets $C(x)$, and put it in the proper format. The prediction sets $C(x)$ can be equivalently represented by a function $C : \mathcal{X} \times \mathcal{Y} \to \{0, 1\}$ such that $C(x, y) = \mathbf{1}[y \in C(x)]$. Furthermore, we can expand the optimization domain from $C(x, y) : \mathcal{X} \times \mathcal{Y} \to \{0, 1\}$ to $C(x, y) : \mathcal{X} \times \mathcal{Y} \to [0, 1]$. One should pay attention that this change does not affect the optimal solutions as the solutions will be integer values. Putting everything together we can write the following problem,

---

**Linear PP**

$$\underset{C(x,y):\mathcal{X} \times \mathcal{Y} \to [0,1]}{\text{Minimize}} \quad \int_{\mathcal{X} \times \mathcal{Y}} C(x, y)p(x)\nu(y)dxdy$$

$$\text{subject to} \quad \mathbb{E}\left[\phi_i(X)\left\{C(X, Y) - (1 - \alpha)\right\}\right] = 0, \quad \forall i \in [1, d].$$

---

Here $\nu$ is the lebesgue measure; recall from Section 2.1 that we defined length through the lebesgue measure on $\mathcal{Y} = \mathbb{R}$, which is equivalent to cardinality of a set in the case where $\mathcal{Y}$ is a discrete and finite set. This problem is a Linear program in terms of $C$ with finitely many constraints. Also, the Linear PP, includes the Primary Problem in the sense that any solution to the Primary Problem is a

feasible solution for Linear PP. That is to say, to prove Proposition 3.1, we just have to prove it for Linear PP.

For $d = 1$, Linear PP can be seen through the lens of the Neyman-Pearson Lemma in Hypothesis Testing [69]. As a result, our analysis of the Linear PP (for general $d$) can be considered as an extension of this lemma which will be done using the KKT conditions.

Now, let us rewrite the Linear PP:

$$\text{Minimize} \quad \int_{\mathcal{X} \times \mathcal{Y}} C(x, y) p(x) \nu(y) dx dy$$

$$\text{subject to:} \quad \int_{\mathcal{X} \times \mathcal{Y}} \phi_i(x) C(x, y) p(x, y) dx dy - (1 - \alpha) = 0, \quad \forall i \in [1, d]$$

$$C(x, y) \in [0, 1] \quad \forall x \in \mathcal{X}, y \in \mathcal{Y}$$

The above is a standard linear program on $C(x, y)$. In what follows, we will find a closed-form solution using the dual of this program. Here, the "optimization variable" $C(x, y)$ belongs to an infinite-dimensional space. Hence, in order to be fully rigorous, we will need to use the duality theory developed for general linear spaces that are not necessarily finite-dimensional. For a reader who is less familiar with infinite-dimensional spaces, what appears below is a direct extension of the duality theory (i.e. writing the Lagrangian) for the usual linear programs in finite-dimensional spaces.

Let $\mathcal{F}$ be the set of all measurable function defined on $\mathcal{X} \times \mathcal{Y}$. Note that $\mathcal{F}$ is a linear space. Let $\Omega$ be the set of all the measurable functions on $\mathcal{X} \times \mathcal{Y}$ which are bounded between 0 and 1; I.e.

$$\Omega = \{C \in \mathcal{F} \text{ s.t. } C : \mathcal{X} \times \mathcal{Y} \to [0, 1]\} \tag{28}$$

Note that $\Omega$ is a convex set. We can then rewrite our linear program as follows:

$$\text{Minimize} \quad \int_{\mathcal{X} \times \mathcal{Y}} C(x, y) p(x) \nu(y) dx dy$$

$$\text{subject to:} \quad \int_{\mathcal{X} \times \mathcal{Y}} \phi_i(x) C(x, y) p(x, y) dx dy - (1 - \alpha) = 0, \quad \forall i \in [1, d]$$

$$C \in \Omega$$

Moreover, let us define the functional $F : \mathcal{F} \to \mathbb{R}$ as

$$F(C) = \int_{\mathcal{X} \times \mathcal{Y}} C(x, y) p(x) \nu(y) dx dy, \tag{29}$$

and also define, for $i \in [1, d]$, the functional $G_i : \mathcal{F} \to \mathbb{R}$ as

$$G_i(C) = \int_{\mathcal{X} \times \mathcal{Y}} \phi_i(x) C(x, y) p(x, y) dx dy - (1 - \alpha). \tag{30}$$

Finally, we define the mapping $\boldsymbol{G} : \mathcal{F} \to \mathbb{R}^d$ as

$$\boldsymbol{G}(C) = [G_1(C), G_2(C), \cdots, G_d(C)].$$

Note that $G$ is a linear (and hence convex) mapping from $\mathcal{F}$ to the Euclidean space $\mathbb{R}^d$.

Using the above-defined notation, our linear program becomes:

$$\text{Minimize} \quad F(C)$$
$$\text{subject to:} \quad \boldsymbol{G}(C) = \boldsymbol{0}$$
$$C \in \Omega$$

where $\boldsymbol{0} \in \mathbb{R}^d$ is the all-zero vector.

Note that the feasibility set of the above program is non-empty, as $C(x, y) = 1 - \alpha$, for all $(x, y) \in \mathcal{X} \times \mathcal{Y}$, is a feasible point. We can now use the duality theory of convex programs in vector spaces (See Theorem 1, Section 8.3 of [68]; Also see Problem 7 in Chapter 8 of the same reference). Specifically, let OPT be the optimal value achievable in the above linear program. Then, there exists a vector $\boldsymbol{\beta} \in \mathbb{R}^d$ such that the following holds:

$$\text{OPT} = \inf_{C \in \Omega} \{F(C) - \langle \boldsymbol{\beta}, \boldsymbol{G}(C) \rangle\}, \tag{31}$$

where $\langle \boldsymbol{\beta}, \boldsymbol{G}(C) \rangle$ denotes the Euclidean inner-product of the two vectors $\boldsymbol{\beta}, \boldsymbol{G}(C) \in \mathbb{R}^d$. Here, note that the vector $\boldsymbol{\beta}$ is the usual Lagrange multiplier.

By denoting $\boldsymbol{\Phi}(x) = (\phi_1(x), \phi_2(x), \cdots, \phi_d(x))$, and using (30), we can write

$$\langle \boldsymbol{\beta}, \boldsymbol{G}(C) \rangle = \int_{\mathcal{X} \times \mathcal{Y}} \langle \boldsymbol{\beta}, \boldsymbol{\Phi}(x) \rangle C(x,y) p(x,y) dxdy - (1-\alpha)\langle \boldsymbol{\beta}, \boldsymbol{1} \rangle,$$

where $\boldsymbol{1} \in \mathbb{R}^d$ is the all-ones vector. As a result, by using (29), in order to solve the optimization in (31) we need to solve the following optimization:

$$\inf_{C \in \Omega} \left\{ \int_{\mathcal{X} \times \mathcal{Y}} C(x,y) \left( p(x)\nu(y) - \langle \boldsymbol{\beta}, \boldsymbol{\Phi}(x) \rangle p(x,y) \right) dxdy \right\} + (1-\alpha)\langle \boldsymbol{\beta}, \boldsymbol{1} \rangle.$$

And by noting the fact the $p(x,y) = p(x)p(y \mid x)$, and removing the term $(1-\alpha)\langle \boldsymbol{\beta}, \boldsymbol{1} \rangle$ which is independent of $C$, our dual optimization becomes:

$$\inf_{C \in \Omega} \left\{ \int_{\mathcal{X} \times \mathcal{Y}} C(x,y) \left( \nu(y) - \langle \boldsymbol{\beta}, \boldsymbol{\Phi}(x) \rangle p(y|x) \right) p(x) dxdy \right\} \tag{32}$$

Now, it is easy to see that as $C(x,y) \in [0,1]$ (due to the constraint $C \in \Omega$), the above optimization problem has a closed-form solution $C^*$:

$$C^*(x,y) = \begin{cases} 1 & \text{if } \langle \boldsymbol{\beta}, \boldsymbol{\Phi}(x) \rangle p(y|x) > \nu(y), \\ 0 & \text{if } \langle \boldsymbol{\beta}, \boldsymbol{\Phi}(x) \rangle p(y|x) < \nu(y), \\ \in \{0,1\} & \text{otherwise.} \end{cases} \tag{33}$$

Finally, note that in this paper the measure $\nu$ is considered to be the Lebesgue measure – see Section 2.1 – hence, up to a constant normalization factor that can be absorbed into $\boldsymbol{\beta}$, we have $\nu(y) = 1$ for all $y \in \mathcal{Y}$.

The structure given in (33) is also sufficient. I.e. for any pair $(\boldsymbol{\beta}, C^*)$ such that (i) $C^*$ has the form given in (33); and (2) $C^*$ satisfies the coverage-constraints $\boldsymbol{G}(C^*) = 0$, then $C^*$ is an optimal solution of the Primary Problem(F). This sufficiency result follows again from the duality theory of linear programs in linear spaces; E.g. see Theorem 1 in Section 8.4 of [68]. This necessary and sufficient condition is known as Strong Duality, which then results in the equivalence of Minimax Problem and Primary Problem, i.e we can change the order of min and max.

# G   Proofs of Section 4

**Proof of Theorem 4.4:** Let us recall the algorithm.

---
**CPL:**
$$\underset{f \in \mathcal{F}}{\text{Minimize}} \; \underset{h \in \mathcal{H}}{\text{Maximize}} \; \tilde{g}_{\alpha,n}(f,h),$$
$$\text{where, } C_h^S(x) = \{ y \in \mathcal{Y} \mid S(x,y) \le h(x) \}.$$

---

For the ease of notation let us call $\tilde{g}(f,h) = \tilde{g}_{\alpha,n}(f,h)$. let us also call the stationary solution to CPL by $h_{\text{CPL}}^*$ and $f_{\text{CPL}}^*$. Now since $\tilde{g}$ is a smooth function with respect to both of its arguments we have the following optimality condition,

$$\left. \frac{d}{d\varepsilon} \tilde{g}(f_{\text{CPL}}^* + \varepsilon f, h_{\text{CPL}}^*) \right|_{\varepsilon=0} = 0, \quad \text{for every } f \in \mathcal{F}. \tag{34}$$

Recall that $\mathcal{F}$ is a $d$-dimensional affine class of functions over the basis $\Phi = [\phi_1, \cdots, \phi_d]$. We can then rewrite 34 as what follows.

$$\left. \frac{d}{d\varepsilon} \tilde{g}(f_{\text{CPL}}^* + \varepsilon\phi_j, h_{\text{CPL}}^*) \right|_{\varepsilon=0} = 0, \quad \text{for every } j \in [1, \cdots, d]. \tag{35}$$

Now looking at the definition of $g(f,h) =$

$$\frac{1}{n} \sum_{i=1}^{n} \left[ f(x_i) \left\{ \tilde{\boldsymbol{1}}[S(x_i, y_i), h(x_i)] - (1-\alpha) \right\} \right] - \frac{1}{n} \sum_{i=1}^{n} \int_{\mathcal{Y}} \left( \tilde{\boldsymbol{1}}[S(x_i, y) \le h(x_i)] - (1-\alpha) \right) dy,$$

We can take the derivative with respect to $f$. Hence we can rewrite 35 as what follows:

$$\frac{1}{n}\sum_{i=1}^{n}\left[\phi_j(x_i)\left\{\tilde{\mathbf{1}}[S(x_i,y_i),h^*_{\mathrm{CPL}}(x_i)]-(1-\alpha)\right\}\right]=0, \quad \text{for every } j \in [1,\cdots,d]. \quad (36)$$

One can think of the mathematical term above as the smoothed version of coverage under covariate shift $\phi_j$. Now we can apply Lemma E.3. Therefore, fixing $j \in [1,\cdots,d]$, with probability $1-\delta$ we have,

$$\left|\frac{1}{n}\sum_{i=1}^{n}\left[\phi_j(x_i)\left\{\tilde{\mathbf{1}}[S(x_i,y_i),h^*(x_i)]-(1-\alpha)\right\}\right]-\mathbb{E}\left[\phi_j(x)\left\{\tilde{\mathbf{1}}[S(x,y),h^*_{\mathrm{CPL}}(x)]-(1-\alpha)\right\}\right]\right|$$

$$\leq \frac{2B\varepsilon}{\sqrt{2\pi}\sigma}+\frac{\sqrt{2}B\sqrt{\ln\left(\frac{2\mathcal{N}(\mathcal{H},d_\infty,\varepsilon)}{\delta}\right)}}{\sqrt{n}} \quad \text{for any } \varepsilon > 0.$$

Combining with (36) we get,

$$\left|\mathbb{E}\left[\phi_j(x)\left\{\tilde{\mathbf{1}}[S(x,y),h^*_{\mathrm{CPL}}(x)]-(1-\alpha)\right\}\right]\right| \leq \frac{2B\varepsilon}{\sqrt{2\pi}\sigma}+\frac{\sqrt{2}B\sqrt{\ln\left(\frac{2\mathcal{N}(\mathcal{H},d_\infty,\varepsilon)}{\delta}\right)}}{\sqrt{n}} \quad \text{for any } \varepsilon > 0.$$

Union bounding over (36) we have with probability $1-\delta$ for any $j \in [1,\cdots,d]$,

$$\left|\mathbb{E}\left[\phi_j(x)\left\{\tilde{\mathbf{1}}[S(x,y),h^*_{\mathrm{CPL}}(x)]-(1-\alpha)\right\}\right]\right| \leq \frac{2B\varepsilon}{\sqrt{2\pi}\sigma}+\frac{\sqrt{2}B\sqrt{\ln\left(\frac{2d\mathcal{N}(\mathcal{H},d_\infty,\varepsilon)}{\delta}\right)}}{\sqrt{n}} \quad \text{for any } \varepsilon > 0. \quad (37)$$

In other words, we proved that the expected smoothed version of coverage is bounded. The last step that we have to take is then to prove that the expected smoothed coverage is actually close to the expected actual coverage. The following claim makes this precise.

**Claim 3** *The following inequality holds for every $j \in [1,\cdots,d]$.*

$$\left|\mathbb{E}\left[\phi_j(x)\left\{\tilde{\mathbf{1}}[S(x,y),h^*_{\mathrm{CPL}}(x)]-(1-\alpha)\right\}\right]-\mathbb{E}\left[\phi_j(x)\left\{\mathbf{1}[S(x,y)\leq h^*_{\mathrm{CPL}}(x)]-(1-\alpha)\right\}\right]\right|$$

$$\leq BL\sigma\sqrt{\frac{2}{\pi}}$$

**Proof.**

$$\left|\mathbb{E}\left[\phi_j(x)\left\{\tilde{\mathbf{1}}[S(x,y),h^*_{\mathrm{CPL}}(x)]-(1-\alpha)\right\}\right]-\mathbb{E}\left[\phi_j(x)\left\{\mathbf{1}[S(x,y)\leq h^*_{\mathrm{CPL}}(x)]-(1-\alpha)\right\}\right]\right|$$

$$=\left|\mathbb{E}\left[\phi_j(x)\tilde{\mathbf{1}}[S(x,y),h^*_{\mathrm{CPL}}(x)]\right]-\mathbb{E}\left[\phi_j(x)\mathbf{1}[S(x,y)\leq h^*_{\mathrm{CPL}}(x)]\right]\right|$$

$$\overset{(a)}{\leq}\mathbb{E}\left[\left|\phi_j(x)\right|\left|\left(\tilde{\mathbf{1}}(S(x,y),h^*_{\mathrm{CPL}}(x))-\mathbf{1}[S(x,y)\leq h^*_{\mathrm{CPL}}(x)]\right)\right|\right]$$

$$\overset{(b)}{\leq}B\mathbb{E}\left[\left|\left(\tilde{\mathbf{1}}(S(x,y),h^*_{\mathrm{CPL}}(x))-\mathbf{1}[S(x,y)\leq h^*_{\mathrm{CPL}}(x)]\right)\right|\right]$$

$$=B\mathbb{E}_X\mathbb{E}_{S|X}\left[\left|\left(\tilde{\mathbf{1}}(S(x,y),h^*_{\mathrm{CPL}}(x))-\mathbf{1}[S(x,y)\leq h^*_{\mathrm{CPL}}(x)]\right)\right|\right]$$

$$\overset{(c)}{\leq}B\mathbb{E}_X L\sigma\sqrt{\frac{2}{\pi}}$$

$$=BL\sigma\sqrt{\frac{\pi}{2}},$$

where (a) comes from triangle inequality, (b) is derived by the definition of $B = \max_{i \in [1, \cdots, d]} \sup_{x \in \mathcal{X}} \Phi_i(x)$, and (c) is followed by assumption 2 and Lemma E.2. Now combining Claim 3 and (37) we have with probability $1 - \delta$,

$$\left| \mathbb{E}\left[ \phi_j(x) \left\{ \mathbf{1}[S(x,y) \leq h^*_{\mathrm{CPL}}(x)] - (1-\alpha) \right\} \right] \right| \leq$$

$$BL\sigma\sqrt{\frac{\pi}{2}} + \frac{2B\varepsilon}{\sqrt{2\pi}\sigma} + \frac{\sqrt{2}B\sqrt{\ln\left(\frac{2d\mathcal{N}(\mathcal{H}, d_\infty, \varepsilon)}{\delta}\right)}}{\sqrt{n}} \quad \text{for any } \varepsilon > 0.$$

Putting $\sigma = \frac{1}{\sqrt{n}}$ and $\varepsilon = \frac{1}{n}$ we have for every $j \in [1, \cdots, d]$,

$$\left| \mathbb{E}\left[ \phi_j(x) \left\{ \mathbf{1}[S(x,y) \leq h^*_{\mathrm{CPL}}(x)] - (1-\alpha) \right\} \right] \right| \leq \frac{\sqrt{2}B\sqrt{\ln\left(\frac{2d\mathcal{N}(\mathcal{H}, d_\infty, \frac{1}{n})}{\delta}\right)} + BL\sqrt{\frac{\pi}{2}} + \frac{2B}{\sqrt{2\pi}}}{\sqrt{n}}$$

Let us also remind that each element $f \in \mathcal{F}$ can be represented by a $\boldsymbol{\beta} \in \mathbb{R}^d$, where we use the notation $f(x) = \langle \boldsymbol{\beta}, \Phi(x) \rangle \equiv f_{\boldsymbol{\beta}}(x)$ (look at section 2.1 for more details). By linearity of the class $\mathcal{F}$ we can conclude with probability $1 - \delta$ for every $f_{\boldsymbol{\beta}} \in \mathcal{F}$,

$$\left| \mathbb{E}\left[ f_{\boldsymbol{\beta}}(x) \left\{ \mathbf{1}[S(x,y) \leq h^*_{\mathrm{CPL}}(x)] - (1-\alpha) \right\} \right] \right| \leq \frac{||\boldsymbol{\beta}||_1 \sqrt{2}B\sqrt{\ln\left(\frac{2d\mathcal{N}(\mathcal{H}, d_\infty, \frac{1}{n})}{\delta}\right)} + ||\boldsymbol{\beta}||_1 BL\sqrt{\frac{\pi}{2}} + ||\boldsymbol{\beta}||_1 \frac{2B}{\sqrt{2\pi}}}{\sqrt{n}}.$$

## H  Marginal Coverage Regression Experiment

In this section we aim at showcasing the ability of CPL in improving length efficiency in designing prediction sets with marginal coverage validity. We compare the performance of our method, CPL, in terms of marginal coverage and length, to various split conformal prediction (SCP) methodologies. Specifically, we compare with: (i) Split Conformal (SC) [22, 44] and Jackknife [45], as the main methods in standard conformal prediction to achieve marginal validity; (ii) Local Split Conformal (Local-SC) [22, 41, 70] and LocalCP [12], as locally adaptive variants of Split Conformal; (iii) Conformalized Quantile Regression (CQR) [35], as a state-of-the-art method for achieving marginal coverage with small set size.

Following the setup from [35], we evaluate performance on 11 real-world regression datasets (see Appendix J) and report the average performance over all of them. Each dataset is standardized, and the response is rescaled by dividing it by its mean absolute value. We split each dataset into training (40%), calibration (40%), and test (20%) sets.

**First Part:** We focus on methods applicable to black-box predictors and conformity score. We compare SC, Jackknife, Local-SC, LocalCP, and CPL using the conformity score $S(x,y) = |y - f(x)|$, where $f$ is a NN with two hidden layers of 64 ReLU neurons, trained on the training set.

**Second Part:** We evaluate CQR, which requires quantile regressors trained on the training set. We also examine our method combined with CQR (i.e. we use the score obtained by CQR), referred to as CQR+CPL. Tables 1 and 2 summarize our results, reporting average performance over 100 random splits of the datasets. All reported numbers have standard deviations below 1 percent.



Table 1: First part

| Method | Length | Coverage |
|---|---|---|
| Split Conformal | 2.16 | 89.92 |
| Jacknife | 1.95 | 89.81 |
| Local-SC | 1.81 | 89.95 |
| LocalCP | 1.73 | 90.09 |
| CPL | **1.68** | **90.11** |

Table 2: Second part

| Method | Length | Coverage |
|---|---|---|
| CQR+Local-SC | 1.59 | 90.15 |
| CQR+LocalCP | 1.48 | 90.01 |
| CQR+SC | 1.40 | 90.05 |
| CQR+Jacknife | 1.32 | 89.78 |
| CQR+CPL | **1.16** | **90.06** |



In Table 1, our method is shown to improve the interval length using a generic conformity score. In Table 2, our method shows strength with sophisticated scores, highlighting that our minimax

procedure significantly enhances length efficiency across various tasks and scoring methods. Our framework's advantages over CQR are: (i) It can use CQR scores to improve length efficiency, and (ii) it can use other scores, including residual scores with pre-trained predictors. This is advantageous when training quantile regressions from scratch is costly, while pre-trained models are accessible.

# I  CIFAR-10 Experiment

We conducted our experiments on the CIFAR-10 dataset, using two distinct training procedures: empirical risk minimization (ERM) with cross-entropy loss, and conformal training (ConfTr by [65]). The neural network architecture employed in all setups was ResNet-32. To evaluate the predictive performance of these models, we aimed to generate prediction sets that achieve a marginal coverage level of 95

For the conformal training procedure, we utilized a batch size of 500 and employed split conformal prediction to compute the prediction set sizes, which is a crucial component of the conformal training approach. We further fine-tuned the hyperparameters using grid search to optimize the setup.

We considered four experimental scenarios: training the model with either ERM or ConfTr, followed by calibration using either split conformal prediction or CPL. The primary goal was to compare the effectiveness of these approaches in terms of both coverage and prediction set efficiency.

In setups involving ConfTr, we further optimized the data split ratios to improve the length efficiency of the prediction sets, which explains the variation in train/calibration/test splits across the different ConfTr setups. For the ERM-based setups, no optimization was performed on the split ratios, as the focus was to emphasize the black-box nature of calibration approaches like CPL and split conformal prediction, which can act as wrappers around the trained models without altering the training procedure.

As shown in the results (see Table 3), CPL combined with ConfTr produces more efficient prediction sets in terms of average length compared to split conformal prediction. This highlights the effectiveness of CPL in improving length efficiency while maintaining the desired coverage level, particularly in conjunction with conformal training.

| Training | Calibration | Coverage | Avg Length | Samples (Train/Calib/Test) | Base Accuracy |
|----------|-------------|----------|------------|----------------------------|---------------|
| ERM | Split Conformal | 0.951 | 2.36 | 40k/10k/10k | 82.6% |
| ERM | CPL | 0.948 | 2.06 | 40k/10k/10k | 82.6% |
| ConfTr | Split Conformal | 0.954 | 2.11 | 45k/5k/10k | 82.3% |
| ConfTr | CPL | 0.947 | 1.94 | 35k/15k/10k | 82.3% |

Table 3: Comparison of different training and calibration methods

# J  Refrences for 11 datasets for Section H

Here we list all the datasets.

- MEPS-19 [71]
- MEPS-20 [72]
- MEPS-21 [72]
- blog feedback (blog-data)[73]
- physicochemical properties of protein tertiary structure (bio) [74]
- bike sharing (bike) [75]
- community and crimes (community) [76]
- Tennessee's student teacher achievement ratio (STAR) [77]
- concrete compressive strength (concrete) [78]
- Facebook comment volume variants one (facebook-1) [79]
- Facebook comment volume variants two (facebook-2) [80]

