# OpenReview forum: "Length Optimization in Conformal Prediction"
_NeurIPS.cc/2024/Conference — NeurIPS 2024 poster_

### Official Review · Reviewer_xcAK · 2024-07-08

**Soundness:** 3
**Presentation:** 2
**Contribution:** 2
**Rating:** 4
**Confidence:** 5

**Summary:**

The paper proposes a new method to improve the efficiency and conditional validity of conformal prediction methods by proposing a minimax optimization problem, whereby the length of prediction intervals is minimized while ensuring (approximate) conditional coverage. The proposed method, conformal prediction with length optimization (CPL), comes with finite-sample guarantees for both the gap to optimal length and conditional coverage. Empirically, CPL is shown to produce narrower prediction intervals than previous methods both with and without distribution shifts.

**Strengths:**

- Conformal prediction is a relevant uncertainty quantification method, and improving its predictive efficiency and robustness to distribution shifts are extremely important problems in the field. The proposed method is, to the best my knowledge, novel and addresses both issues in a principled manner.
- The proposed method, CPL, can be applied on top of any pre-trained classifier, which facilitates its use in practice.
- CPL comes with convergence guarantees in terms of coverage and predictive efficiency, and the mathematical results seem sound.
- The empirical results show the proposed method outperforms the baselines in terms of predictive efficiency.

**Weaknesses:**

- While the paper is not exactly difficult to follow, I feel the presentation could be greatly improved. Some of the notation is confusing (see e.g. question 3 below) and there is barely any discussion on the theoretical and empirical results (see comments on experiments below).
- I am not entirely convinced of the impact of the contributions of the paper. The novelty is somewhat limited—one could summarize the contribution as applying conformal training ideas [2, 3, 4, 5] on top of the machinery introduced by Gibbs et al. [1]—and this is not compensated by the results, which, at least in the way they are currently presented, are underwhelming: one gets some convergence guarantees, but it is not clear how tight or useful they are,  and one gets some gains in predictive efficiency, but the empirical evaluation leaves a lot to be desired (see comments below).
- Experiments are not convincing.
    - The experiments are not well described or analyzed. There is no description of the setup (only references to other papers). Tables and figures are not properly captioned. There is no report of any statistics besides mean results, despite the authors’ claims in the checklist. The only figure with error bars is Figure 3, but even there the reader has no idea how those were computed and what they mean.
    - I am not convinced the experiments in the paper qualify as “extensive empirical evaluations”. The experiments are very small in scale, and similar methods that also directly optimize for small prediction intervals are missing [2, 3, 4, 5]. Split localized conformal prediction [6] is also a relevant baseline in my opinion, since $h(x)$ is essentially an estimator of the quantile of the conditional distribution of scores given X, as proposed in [6].
    - Additional computational costs incurred by the proposed optimization procedure are not commented on. The computational efficiency of conformal prediction is one of its main selling points, especially with ever larger and more expensive machine learning models, and the cost of any additional step might be relevant. The fact that the experiments in the paper are very small in scale and can be run on CPUs does not mean the computational cost is irrelevant.

### References
[1] Gibbs, Isaac, John J. Cherian, and Emmanuel J. Candès. "Conformal prediction with conditional guarantees." arXiv preprint arXiv:2305.12616 (2023).

[2] Stutz, David, et al. "Learning Optimal Conformal Classifiers." International Conference on Learning Representations.

[3] Yu Bai, Song Mei, Huan Wang, Yingbo Zhou, and Caiming Xiong. Efficient and differentiable conformal prediction with general function classes. arXiv preprint arXiv:2202.11091, 2022.

[4] Bellotti, Anthony. "Optimized conformal classification using gradient descent approximation." arXiv preprint arXiv:2105.11255 (2021).

[5] Correia, Alvaro HC, et al. "An Information Theoretic Perspective on Conformal Prediction." arXiv preprint arXiv:2405.02140 (2024).

[6] Han, Xing, et al. "Split localized conformal prediction." arXiv preprint arXiv:2206.13092 (2022).

### Minor Issues
- I find the usage of the term “structured prediction sets” somewhat unnecessary. At the end of the day, this is no different from group conformal prediction or even adaptive prediction sets, where the threshold depends on the input x. The extra terminology only adds confusion.
- Figure 1 does not add much to the paper, in my opinion.
- Line 7: I believe the hyphen is unnecessary in “Conformal Prediction with Length-Optimization”.
- Line 76: “the” is repeated.
- Line 90: It seems a word is missing after “overly”.
- Line 137: I think “converge” should be “coverage” instead.
- Line 192: “that” should probably be removed.
- Line 228: “relax” should probably be “relaxed”.
- Line 255: Typo “velow”.
- Line 548: Typo $Z{h_2}$
- Line 664: Typo “traingle”

**Questions:**

1. Which data is used to optimize $h(\cdot)$ and $f(\cdot)$ in Algorithm 1? I assume one needs a separate dataset (distinct from the calibration one) so as not to break exchangeability between calibration and test data, but, unless I missed it, this is not mentioned anywhere.
2. In Figure 3, right-hand-side plot, CPL achieves smaller mean interval length than the optimal oracle in many cases. How is that possible?
3. I am somewhat confused by the notation. Sometimes $f(\cdot)$ is used to denote the shift, sometimes it is used to denote the machine learning model. In line 280, what exactly is a NN with two hidden layers?
4. How are $h(\cdot)$ and $f(\cdot)$ parametrized in the experiments? Could the improvement in efficiency provided by CPL in these datasets be explained by the extra learnable parameters in $h(\cdot)$ and $f(\cdot)$? Or in other words, could we get the same performance simply by considering a more powerful model class for the underlying regressor or classifier?
5. Could the authors share some intuition or results on how tight the bounds on Theorem 4.2. and 4.3. are? How do they compare to previous results? For instance, unless I am missing something, in comparison to Theorem 4.2. in this paper, Theorem 2 in [1] seems to provide a tighter bound that converges faster at $O(1/n)$.
6. The description of the covariate shift experiments in section 5.3. is very unclear. Could the authors elaborate further on the experimental setup?

### References
[1] Gibbs, Isaac, John J. Cherian, and Emmanuel J. Candès. "Conformal prediction with conditional guarantees." arXiv preprint arXiv:2305.12616 (2023).

**Limitations:**

The paper does comment on its limitations and provides promising avenues for future work.

---

> ### Author Rebuttal · Authors · 2024-08-04
>
> We thank the reviewer for their detailed feedback. We appreciate the recognition of the strengths of our proposed method, CPL, particularly its novel approach to enhancing predictive efficiency and robustness in conformal prediction. We are glad that the reviewer mentions CPL's potential to be applied on top of any pre-trained classifier, its sound mathematical underpinnings, and its promising empirical results.
>
> Due to space constraints, we will respond to reviewer's concern about novelty and connection to conformal training ideas here, and we have three additional official comments in which we respond to reviewer's concern about the experiments and the other questions.
>
> **Novelty and Fundamental Difference with Conformal Training Ideas:**
> We respond to this comment from three angles. We will also add a discussion in our related work section of the paper with the relevant literature.
>
> **1. Different ideas and approach:**
> Our approach to conformal prediction is from the perspective of uncertainty quantification for a **given black-box model**, without altering the model itself. In the paper, we start by fundamentally characterizing the interplay between conditional validity and length efficiency in a minimax framework (please see Proposition 3.1). In particular, we derive **the optimal** prediction sets using a novel **level set formulation**. Obtaining the optimal sets amount to solving an **equivalent miniamx problem** where the max part guarantees length optimality and the min guarantees coverage. Having the black-box approach in mind, we then relax our minimax approach to search for prediction sets of the form $S(x,y)\leq h(x)$ by fixing the choice of $S$ and **optimizing over the right-hand side**, i.e. the function  $h(x)$. Now, two points are in order. (i) Unlike Gibbs et al. [1], here the covariate adaptivity of the threshold $h(x)$, also roots in **learning features for length optimization**. I.e., we choose an adaptive threshold even for marginal coverage case (and **we show that an adaptive threshold is fundamentally necessary for optimizing length** even in the marginal case). (ii) Conformal training methods aim to produce tight prediction sets by going beyond the black-box approach and altering the base model, i.e. **optimizing the left hand side** of $S(x,y)\leq h$ (while calibrating the constant $h$). This should clear the role of Figure 1 of our submission, where conformal training ideas belong to the first box on the left and CPL belongs to the right end.
>
> Therefore, we believe our approach **is not an extension of conformal training ideas** to Gibbs et al. It has new insights (level set perspective) and new techniques (minimax length optimization by threshold adaptivity). We now illustrate these points in depth with a simple example.
>
> **2. Simple Example:**
> We start with the example on the second page of our submission. We focus exclusively on marginal coverage (in this case, Gibbs et al. [1] reduces to split conformal). Analogous to extending conformal training ideas, one might attempt to improve interval length by optimizing $\hat{f}$ in $|y-\hat{f}(x)|\leq q$. However, in our framework, we keep $\hat{f}$ fixed and instead optimize $h$ in $|y-\hat{f}(x)|\leq h(x)$. We encourage the reviewer to look at the details of the example and see the **necessity of the threshold adaptivity** to reduce interval length regardless of the choice of $\hat{f}$ (even when $\hat{f}(x) = E[Y|X=x]$). This necessity is due to the variance structure of the noise, which can't be captured by the base model alone and **requires an adaptive threshold**.
>
> **3. Additional Experiment:**
> **CPL and conformal training can also be used in conjunction** to enhance prediction set efficiency. To illustrate this, we conducted an experiment on CIFAR-10. In this experiment, we explored four different scenarios where the base model is either trained using simple Empirical Risk Minimization (ERM) with cross-entropy loss or using conformal training (confTR). The calibration phase is then performed using either the split conformal method or CPL. In this setup, we focus solely on marginal coverage, setting the nominal coverage level to 0.95.
> | **Training** | **Calibration** | **Coverage** | **Avg Length** | **Samples (Train/Calib/Test)** | **Base Accuracy** |
> |--------------|-----------------|--------------|----------------|-------------------------------|-------------------|
> | ERM          | Split Conformal | 0.951        | 2.36           | 40k/10k/10k                   | 82.6%             |
> | ERM          | CPL             | 0.948        | 2.06           | 40k/10k/10k                   | 82.6%             |
> | confTR       | Split Conformal | 0.954        | 2.11           | 45k/5k/10k                    | 82.3%             |
> | confTR       | CPL             | 0.947        | 1.94           | 35k/15k/10k                   | 81.8%             |
>
> The results demonstrate that confTR can be further improved by using CPL as the calibration method. This supports our main message: even when the model is fixed, there is still potential to optimize length efficiency through a minimax framework that refines the threshold.
>
> For ConfTR, as suggested by the original ConfTR paper, we first train the ResNet34 by ERM using cross entropy and then fine-tune it with confTR. For the inner maximization ($h(x)$) of CPL we fine-tuned only one additional (last) layer of the model learned in the training time. Hence, the CPL calibration step is significantly lighter in computation with respect to train step with either ERM or confTR. For confTR, we also optimized the train/calibration split to achieve maximum length reduction. However, for the ERM scenarios, we did not optimize this split, bolding our black-box perspective. We used APS scoring for calibration. The reported numbers are averaged over 20 independent random splits of the data. The standard deviation errors for all the reported coverage and length numbers are below 0.02.

---

> ### Author Response · Authors · 2024-08-06
> **Addressing Reviewer's Comments on the Experiments**
>
> **Scaleability of Experimental Design**
>
> In our section on the experiments, we evaluate the performance of CPL in three different setups ranging in both regression and classification settings on real-world and synthetic datasets. Our experiments aim to compare CPL with other **state-of-the-art conformal prediction algorithms that treat the base model as a black box**. All three experimental setups are standard and have been used in the community to measure the performance of CP methods. In particular, section 5.1 setup is standard in comparing methods that provide marginal coverage as introduced by [2], section 5.2 is standard for group conditional methods as designed by [3], and finally, 5.3 setup is identical to that used by Gibbs et al. [1] to showcase performance under a class of covariate shifts.
>
> Also, in response to the reviewers request, we will add the method of "Split localized conformal prediction" to the baselines of section 5.1.
>
> **Additional Experiment on LLM Question Answering**
>
> To further demonstrate the applicability/scalability of our method, we will include a large-scale experiment involving LLM question-answering in the camera-ready version. This experiment uses multiple-choice question-answering datasets, including TruthfulQA, MMLU, OpenBookQA, PIQA, and BigBench. The calibration data for MMLU alone consists of approximately 100,000 prompts and answers, illustrating the large scale of this experiment.
>
> The goal is to quantify the uncertainty of Llama 2 and create prediction sets using this model. We follow a procedure similar to that proposed by prior work [4], adapting it as follows: for each dataset, the task is a multiple-choice question-answering. We pass the question to Llama 2 using a consistent prompt: "This is a 4-choice question that you should answer: {question} The correct answer to this question is:" We examine the logits of the first output token for options A, B, C, and D. Applying a softmax function gives us a probability vector, and we define the conformity score as $1 − \text{probability of the correct option}$
> similar to $1-f(x)_y$ for classification.
>
> CPL is implemented using a linear head (as $h(x)$) on top of a pre-trained GPT-2. In more detail, with GPT-2 having 768-dimensional hidden layers, the inner maximization involves optimizing a 768-dimensional linear map from GPT-2’s last hidden layer representations to a scalar. We also implemented the baseline method that directly applies the split conformal method to the scores. Please **see the uploaded PDF for plots** and a caption for more details.
>
> **Computational Cost**
>
> We acknowledge and agree with the reviewer's comment on the importance of computational cost. Two points about this matter are in order. (i) Going beyond the simple solution of split conformal for black box base models, would require an adaptive threshold (as argued in our paper). This adaptive threshold is often achieved by an extra optimization on the calibration data (e.g. see [1] and [2]). We are introducing length optimization on top of conditional validity, hence it is expected that one would have to still do optimization to find the threshold. (ii) our computational cost is not more than training an ML model for the threshold, and it can effectively be solved by gradient descent. For example, computing the gradients for inner maximization amounts to computing the gradient with respect to the parameters of a neural network (hence is totally scalable). Also, the outer minimization step is computationally lightweight and can be viewed as a form of hyper-parameter tuning. For instance, in scenarios focused on marginal coverage, the outer minimization reduces to a simple scalar optimization. We will ensure to include a comprehensive discussion on scalability in the camera-ready version of the paper.
>
> **Description of the Experiments**
>
> We acknowledge the reviewer's comment on the importance of a detailed description and detailed report of the results. The brevity in some parts is due to space constraints. In the revised version, we will take advantage of the allowed extra page to include more details on the setups and plot captions.
>
> We also acknowledge the importance of higher-order statistics. Some of these reports are in the paper (e.g. see lines 283-284 or 297), and some others were dropped as the error bars were negligible. We will make sure to improve the quality of statistics reports in the revised manuscript. We thank the reviewer for this comment.
>
> We will also provide full details of implementation specifications, and include a link to our Python implementations in the camera-ready version.
>
> **References**
>
> [1] Gibbs, Isaac, et al.. "Conformal prediction with conditional guarantees." (2023).
>
> [2] Romano, Y., et al. "Conformalized quantile regression" (2019).
>
> [3] Jung, Christopher, et al. "Batch multivalid conformal prediction." (2022).
>
> [4] Kumar, Bhawesh, et al. "Conformal prediction with large language models ...." (2023).

---

> ### Author Response · Authors · 2024-08-06
> **Response to Individual Questions**
>
> “Which data is used to optimize...”
>
> The algorithm 1 uses all the calibration data to optimize both $f(\cdot)$ and $h(\cdot)$, through a minimax procedure, where we alternatively take gradient descent with respect to $f(\cdot)$ and gradient ascent with respect to $h(\cdot)$. In that process, the outer min ensures the conditional validity of the prediction sets and the inner max improves length efficiency. We will clarify this in the updated manuscript.
>
> “In Figure 3, right-hand-side plot, CPL achieves smaller ...”
>
> The optimal oracle has the smallest interval length **averaged over all the covariates (all the test samples)**, while achieving valid coverage with respect to each of the groups. You can see this by looking at the first bars on the right of the “Figure 3, right-hand-side plot”. The optimal oracle is achieving mean length interval of around 8.0 while CPL is clearly achieving more than 8. However, looking at the mean interval conditioned on each grouping of the data (which corresponds to the rest of the bars in the “Figure 3, right-hand-side plot”) CPL has smaller or larger mean interval lengths in different groups. We will elaborate on this matter in the revised version.
>
> “I am somewhat confused by the notation. Sometimes ...”
>
> We acknowledge the difficulty in the notation and we will make sure to improve the clarity of the notations in the revised version. In general, throughout the paper $\mathcal{F}$ denotes the class of covariate shift and $f\in\mathcal{F}$ a single covariate shift function in the class. However, sometimes $f(\cdot)$ is also used to address the base model (which will be fixed).
> In line 280, by a NN with two hidden layers we meant our neural network architecture consists of three fully connected layers, with ReLU nonlinearities between layers. The first layer takes as input the feature vector X and outputs 64 hidden variables. The second layer follows the same template, outputting another 64 hidden variables. Finally, a linear output layer returns a pointwise
> estimate of the response variable Y.
>
> "How are $f(.)$ and $h(.)$ are parametrized in the experiments? Could the ..."
>
> The function $h(\cdot)$ is parameterized by the natural parameters of the machine learning model used to implement it. For example, $h(\cdot)$ could be implemented as a neural network (NN). In each experimental setup, we specify the ML model we use for $h(\cdot)$. In our experiment setups, simple MLP models or fine-tuning the last layer of a pre-trained model suffices to get a good performance.
>
> For the function $f(\cdot)$, recall that we define $\mathcal{F} = \{\langle \boldsymbol{\beta}, \Phi(x)\rangle \mid \boldsymbol{\beta} \in \mathbb{R}^d\}$, where each $f \in \mathcal{F}$ can be represented as $f(x) = \langle \boldsymbol{\beta}, \Phi(x)\rangle \equiv f_{\boldsymbol{\beta}}(x)$. Therefore, iterating on $f$ can be implemented by updating $\boldsymbol{\beta} \in \mathbb{R}^d$. For instance, in the case of marginal coverage, $\boldsymbol{\beta} \in \mathbb{R}$, hence it becomes a scalar optimization. For the case of conditional coverage with respect to $m$ groups, $\boldsymbol{\beta}$ would be an $m$-dimensional vector (see section 2.1).
>
> In a nutshell, the number of parameters used for $h(\cdot)$ and $f(\cdot)$ is typically significantly less than the number of (learned) parameters at training time.
>
> Concerning a more powerful base model, as mentioned above, we treat the base model as a black-box and quantify its uncertainty. We have also added a CIFAR-10 experiment (described earlier) to showcase that CPL can also improve the length on top of training methods like conformal training.
>
> "Could the authors share some intuition or results on how tight..."
>
> The coverage results in conformal prediction are typically provided in two ways. One is over the expectation on the calibration data (e.g. the one in Gibbs et al. [1]). The others are PAC-style guarantees similar to what we provide in Theorems 4.2. and 4.3. PAC-style guarantees are very common in the CP literature (e.g. see [2], [3], [4]). The PAC-style guarantees are stronger in the sense that they are stated with high probability over the draw of the calibration (rather than over expectation). These bounds are also known as training-conditional guarantees in the literature. PAC-style guarantees of $O(1/\sqrt{n})$ are generally optimal (e.g. see [2], [3]).
>
> "The description of the covariate shift experiments in"
>
> We will respond to this question fully in the next official comment.
>
> **Regarding the minor comments, we thank the reviewer very much and we'll revise the text accordingly.**
>
> **References**
>
> [1] Gibbs, Isaac, et al.. "Conformal prediction with conditional guarantees." (2023).
>
> [2] Sangdon Park, et al. "Pac confidence sets for deep neural networks via calibrated prediction." 2019.
>
> [3] Vovk, Vladimir. "Conditional validity of inductive conformal predictors." 2012.
>
> [4] Jung, Christopher, et al. "Batch multivalid conformal prediction." (2022).

---

> ### Author Response · Authors · 2024-08-06
> **Experimental Setup in Section 5.3**
>
> Here we respond to the reviewer's concern regarding the clarity of our description of the experimental setup in Section 5.3. Below, we provide a more detailed explanation of our experimental setup for the RxRx1 dataset from the WILDS repository, highlighting key aspects and methodologies.
>
> Dataset Overview:
>
> The RxRx1 dataset consists of cell images obtained through fluorescent microscopy, with the task of predicting one of 1,339 genetic treatments. These images come from 51 different experiments, each representing covariate shifts due to varying execution and environmental conditions. Our goal is to create prediction sets that maintain valid coverage across these shifts, a challenge due to heterogeneity in the quality of predictions across different experiments and cell types.
>
> Experimental Design:
>
> 1. **Covariate Shift Characterization:**
>
>    - We follow a data-driven approach to characterize covariate shifts, similar to the method in Gibbs et al. (2023), by splitting the calibration data in half and estimate the probabilities of experiment membership in the first half. We estimate these probabilities using $\ell_2$-regularized multinomial linear regression on the pre-trained ResNet50 model's feature map outputs. We then use these estimated probabilities to form the class of covariate shifts.
>
> 2. **Model and Data Splitting:**
>
>    - We use a ResNet50 as the base model, pre-trained on 37 of the 51 experiments. These 37 experiments contain a total of approximately 91000 images. The remaining 14 experiments are split into calibration and test sets, each of which with approximately 16000 images. For CPL inner maximization (corresponds to $h(.)$), we train a linear head on the last-layer representations of the pre-trained ResNet50 model.
>
> 3. **Conformity Scores and Temperature Scaling:**
>
>    - For each image $x$, the ResNet50 model outputs weights $\{f^i(x)\}_{i=1}^{1339}$ for the treatments. These are converted to probability weights $\pi^i(x)$ using softmax with a temperature parameter $T$ that is calibrated to adjust these probabilities. Temperature refines the accuracy of these probabilities. The conformity score $S(x, y)$ is computed as the sum of probabilities for treatments where the predicted probability $\pi^i(x)$ exceeds that of the correct treatment $\pi_y(x)$.
>
> 4. **Comparative Analysis:**
>
>    - We compare CPL with the Conditional Calibration algorithm, as proposed by Gibbs et al. (2023), and the Split Conformal method. Our findings, depicted in Figure 4, show that while both CPL and Conditional Calibration maintain almost valid coverage across shifts, CPL significantly reduces the average prediction set size due to its length optimization, which enhances the efficiency of prediction sets.
>
> We will take advantage of the extra page in the revised manuscript to enhance the description of section 5.3.

---

> > ### Comment · Reviewer_xcAK · 2024-08-13
> >
> > I thank the authors for the extensive rebuttal and extra experimental results. I am inclined to maintain my score, as the current version of the paper would need a thorough revision to include all the clarifications and new experimental details and results. That said, I am not going to oppose acceptance if the other reviewers are in favor of it.
> >
> > Further, I would still argue the main contribution lies in combining conformal training (or length optimization) with the framework of Gibbs et al., and some of the distinctions made by the authors with respect to related work are somewhat arbitrary. For instance, emphasizing that the method works with black-box predictors is not helpful, since conformal training can also be applied on top of a black-box model by training an additional layer, like Stutz et al. did in some of their experiments. The distinction between manipulating the score function or the threshold is also not clear cut, since one could argue that $h(x)$ could be absorbed in $s(x, y)$. That is not to say the contribution of the paper is not valid. It is. Though I would argue the contribution is not strong enough to outweigh the other problems with the presentation and experimentation in the current version of the paper.
> >
> > As a final question, in the additional experiments in the comparison against conformal training, why are the train/calibration/test splits not the same in the last two rows?

---

> ### Author Response · Authors · 2024-08-13
>
> Dear Reviewer,
>
> First of all, we’d like to thank you for your thoughtful feedback and for considering our rebuttal and additional experimental results! We appreciate your acknowledgement of CPL’s contribution to enhancing predictive efficiency and robustness in conformal prediction. We agree with you regarding the need to incorporate these additional details which we will make sure to add in the revised version (given the extra page). And we hope that there is more opportunity to discuss these points, as we think that this feedback has improved our paper.
>
> Regarding the importance of the black-box approach, the fundamental objective is to quantify the uncertainty of a model as it is used for point prediction. This means taking the model as given and focusing on understanding its uncertainty behavior without altering its internal parameters. In contrast, if an additional layer is trained on top of a model, as seen in some experiments by Stutz et al., it effectively quantifies the uncertainty of a different model. I.e., we would like to keep the model unchanged and provide uncertainty sets around what the model is predicting (i.e., quantify the uncertainty of the model). If we fine-tune the features learned by the model (e.g. by using some extra layers), the fine-tuned version will be a different model, and the resulting prediction sets quantify the uncertainty of the fine-tuned model (i.e. they are not a wrapper around the original model). This distinction is central to the standard approach in the conformal prediction literature.
>
> To help clarify that our arguments are not “arbitrary”, let us mention two quotes from Gibbs et al. highlight the black-box perspective: (1) “In the predictive inference literature, conformal inference is often described as a protective layer that lies **on top of a black-box machine learning model** and transforms its point predictions into valid prediction sets,” and (2) “Emulating split conformal, we design our procedure as a **wrapper that takes any black-box machine learning model** as input. We then compute conformity scores that measure the accuracy of **this model’s predictions** on new test points.” These statements underscore the significance of treating the model as a black box.
>
> That said, we do not believe that CPL and conformal training ideas should be seen as rivals but as complementary methods applicable at different stages of the conformal prediction (CP) pipeline. The CIFAR-10 experiment in our rebuttal demonstrates how both methods can be used together to optimize prediction set efficiency.
>
> To further elaborate on scenarios where CPL is more applicable than conformal training due to its black-box approach: In many real-world applications, direct access to a model’s internal parameters is restricted due to privacy, security, or proprietary concerns. For instance, models like GPT-4 are closed-source, making it crucial to quantify uncertainty without altering the model itself. Even if we had access, altering the model to improve length efficiency could potentially degrade both in-distribution and out-of-distribution performance of the base model. Moreover, keeping the base model for point prediction and a fine-tuned one for constructing prediction sets means quantifying the uncertainty of a different predictive model, which is not aligned with our objective.
>
> Additionally, looking beyond the important black-box distinction, our example in the introduction of the paper illustrates that the absorption of the threshold into the score ($|y - f(x)|$), as suggested by the reviewer, is not sufficient to achieve the optimal length in general. In that example (which we also highlighted in our rebuttal), we have argued that for length optimization, one must optimize the threshold, and this cannot be replaced by further optimizing the base model. I.e., **the threshold $h(x)$ can not be absorbed into $S(x,y)$**. The variance structure of the noise requires an adaptive threshold for length optimization, which is not possible through adjustments to the base model alone if we want to achieve optimal length.
>
> We hope this response clarifies the distinctions and importance of the black-box approach in our work and further elaborates on the complementary nature of CPL and conformal training.
>
> Regarding the additional question, for the cases involving confTR, we also optimized over the split ratios between calibration and training to maximize length efficiency. I.e., since the underlying model is optimized in ConfTR, then we can also treat the split between training and collaboration data as a hyper-parameter that is optimized.  However, for the cases of ERM we used only 10k calibration and did not optimize over the split, to highlight the black box perspective. I.e,. in the black-box approach most of the training data is often used for training the model (to maximize its accuracy), and the prediction sets are constructed afterwards using a relatively small set of calibration data.

---

### Official Review · Reviewer_5vGW · 2024-07-10

**Soundness:** 3
**Presentation:** 3
**Contribution:** 3
**Rating:** 6
**Confidence:** 4

**Summary:**

The paper presents a novel framework for conformal prediction that aims to balance conditional validity and length efficiency. The authors propose CPL to address the challenges of constructing prediction sets that are conditionally valid and have optimal length. This paper is well-written, provides a comprehensive review of related work, and demonstrates significant advancements in the field of conformal prediction.

**Strengths:**

1) The introduction of CPL is a significant contribution. It fills a gap in the existing literature by providing a unified approach to achieving both conditional validity and length efficiency.

2) The paper provides both infinite sample and finite sample guarantees for the proposed method. This adds a strong theoretical foundation to the practical applicability of the method.

3) The authors conducted extensive experiments on diverse real-world and synthetic datasets across both classification and regression settings. The results demonstrate the superior performance of CPL compared to state-of-the-art methods.

4) The use of a minimax framework to derive optimal prediction sets is a sophisticated approach that shows deep theoretical insights.

5) CPL can use any conformity score and adapt to different coverage requirements, making it highly versatile.

**Weaknesses:**

1) The paper assumes L-Lipschitz continuity for conditional distributions and bounded derivatives for conformity scores. While these assumptions are standard, they might limit the applicability of CPL in scenarios where these conditions do not hold.

2) The computational complexity of the proposed method, especially the inner maximization and outer minimization steps, is not thoroughly discussed. Addressing the scalability of CPL for very large datasets would be beneficial.

3) While the authors acknowledge the limitation regarding handling infinite-dimensional classes, they do not provide a concrete roadmap for future work in this area. A more detailed discussion on potential extensions and their feasibility would strengthen the paper.

**Questions:**

1) The paper assumes L-Lipschitz continuity for conditional distributions and bounded derivatives for conformity scores. How sensitive is the proposed CPL framework to violations of these assumptions in practical applications?

2) The proposed CPL method involves both inner maximization and outer minimization steps. What is the computational complexity of these steps, and how does it scale with the size of the dataset?

3) The paper discusses handling various classes of covariate shifts. How robust is the CPL method to unexpected or unmodeled shifts in the data distribution?

4) How does the performance of CPL vary with different choices of conformity scores and hypothesis classes H? Can the authors provide a sensitivity analysis to show the robustness of their method?

**Limitations:**

yes

---

> ### Author Rebuttal · Authors · 2024-08-05
>
> We would like to express our sincere gratitude to the reviewer for their thoughtful and detailed evaluation of our submission. We are glad that the introduction of the Conditional Prediction Length (CPL) framework has been recognized as a significant contribution to the field of conformal prediction, and we appreciate the acknowledgment of our theoretical foundations and extensive experimental validation.
>
> **Regarding Assumptions**
>
> While the Lipschitz continuity and bounded derivative assumptions are necessary for the theoretical aspects of our work, they are not prerequisites for running CPL. The CPL framework is designed to be agnostic to these assumptions. In our experiments, we demonstrate CPL's robust performance using various practical score functions, including $1-f(x)_y$ for classification, which may inherently violate these assumptions due to its discrete nature. Also in regression tasks, we have showcased CPL can be used in conjunction with practical scores like CQR and improve the length efficiency. It is worth noting that other methods, such as split conformal prediction, also require similar assumptions for meaningful analysis. For example, if the score distributions include deltas (violating the Lipschitz assumption), split conformal methods might produce overly conservative prediction sets.
>
> **Scalability**
>
> Solving minimax problems is a common approach within the machine learning community, particularly in areas such as adversarial learning. In the specific case of CPL, the outer minimization step is computationally lightweight and can be viewed as a form of hyperparameter tuning. For instance, in scenarios focused on marginal coverage, the outer minimization reduces to a simple scalar optimization. Additionally, the outer minimization is linear in terms of its variables, which simplifies gradient calculations. The complexity of the inner maximization is comparable to training a standard machine learning model; e.g. by using standard gradient steps on the parameters. In general, the complexity of the inner maximization is similar to complexity of a neural network; and CPL requires very simple neural networks--e.g. two layer MLPs, or by fine-tuning a pre-trained model. We will ensure to include a comprehensive discussion on scalability in the camera-ready version of the paper.
>
> **Infinite Dimensional Classes of Covariate Shifts**
>
> Addressing infinite-dimensional covariate shifts for coverage validity is an exciting direction for future work. A promising approach involves employing regularization techniques, as discussed by Gibbs et al.[1], though their framework lacks length optimization. Exploring regularization methods that integrate length optimization while maintaining strong duality results, as in Proposition 3.1, presents a valuable research avenue. We will expand on this idea in the future work section.
>
> **Unexpected or Un-modeled Covariate Shifts**
>
> As the reviewer noted, our framework presumes a pre-defined class of covariate shifts. Addressing unexpected or un-modeled shifts involves several considerations: (i) Providing valid coverage without any prior knowledge or assumptions about covariate shifts is equivalent to solving the full conditional coverage problem, which is infeasible (as discussed in [1]). Thus, some assumptions or prior knowledge are essential. (ii) There is a growing body of work on leveraging unlabeled data from target distributions to ensure valid coverage amidst unknown covariate shifts. Investigating length optimization in such contexts could be a fruitful research direction. (iii) Utilizing a pre-trained foundational model (fine-tuned for the task) to define the class of covariate shifts can potentially capture unexpected shifts, similar to the experimental setup in section 5.3, where covariate shifts were not explicitly defined.
>
> **Regarding the Hypothesis Class $H$**
>
> Our experiments explore a variety of practical conformity scores in both regression and classification settings, demonstrating CPL's adaptability and robust performance across different scenarios. We have also evaluated CPL's effectiveness using both simple models, like Multi-Layer Perceptrons (MLPs), and by fine-tuning more complex models. If the model is too simple, it may fail to capture the necessary features for effective length optimization. Conversely, overly complex models might lead to overfitting, which could hinder finite sample generalization.
>
> To provide a sensitivity analysis of $H$ **we conducted a numerical evaluation** based on the experimental setup in Section 5.2. We report the mean interval length over the test data while varying the number of activation functions in the hidden layer of the two-layer MLP used as $h(.)$ and the number of calibration samples and test samples are fixed to 50K. The results, depicted in the **uploaded PDF**, demonstrate how the mean interval length changes with different MLP complexities, highlighting CPL's robust performance across a range of model complexities.
>
> [1] Gibbs, Isaac, John J. Cherian, and Emmanuel J. Candès. "Conformal prediction with conditional guarantees." arXiv preprint arXiv:2305.12616 (2023).

---

> ### Author Response · Authors · 2024-08-06
> **Additional Experiment on LLM Question Answering**
>
> To further demonstrate the applicability/scalability of our method, we will include a large-scale experiment involving LLM question-answering in the camera-ready version. This experiment uses multiple-choice question-answering datasets, including TruthfulQA, MMLU, OpenBookQA, PIQA, and BigBench. The calibration data for MMLU alone consists of approximately 100,000 prompts and answers, illustrating the large scale of this experiment.
>
> The goal is to quantify the uncertainty of Llama 2 and create prediction sets using this model. We follow a procedure similar to that proposed by prior work [4], adapting it as follows: for each dataset, the task is a multiple-choice question-answering. We pass the question to Llama 2 using a consistent prompt: "This is a 4-choice question that you should answer: {question} The correct answer to this question is:" We examine the logits of the first output token for options A, B, C, and D. Applying a softmax function gives us a probability vector, and we define the conformity score as $1 − \text{probability of the correct option}$
> similar to $1-f(x)_y$ for classification.
>
> CPL is implemented using a linear head (as $h(x)$) on top of a pre-trained GPT-2. In more detail, with GPT-2 having 768-dimensional hidden layers, the inner maximization involves optimizing a 768-dimensional linear map from GPT-2’s last hidden layer representations to a scalar. We also implemented the baseline method that directly applies the split conformal method to the scores. Please **see the uploaded PDF for plots** and a caption for more details.

---

> > ### Comment · Reviewer_5vGW · 2024-08-12
> >
> > I thank the authors for the detailed response. I maintain my positive score of weak accept.

---

> > > ### Author Response · Authors · 2024-08-12
> > >
> > > We would like to express our gratitude to the reviewer for their thoughtful review and positive assessment of the paper. We appreciate the valuable feedback, which has helped to improve our work.

---

### Official Review · Reviewer_qXD6 · 2024-07-13

**Soundness:** 3
**Presentation:** 3
**Contribution:** 3
**Rating:** 7
**Confidence:** 3

**Summary:**

This paper introduces a new formulation of conformal prediction that not only aims to achieve the coverage property but also explicitly optimises the length of the prediction intervals. The framework is generic and can be applied to various conformity scores. The authors evaluate the approach through extensive empirical evaluations.

**Strengths:**

* The paper is well-written and enjoyable to read.
* The motivating example and results, particularly the finding that the optimal length of the predictive set should be the level-set of the conditional probability and the smoothing step, are insightful and valuable for the community.
* The experiments are thorough and cover different dimensions, including marginal coverage, group-conditional coverage, and performance under covariate shift.

**Weaknesses:**

* The readability of the figures could be improved.
* The notation used for expectations, such as in Equation 2, is a bit misleading. It appears as though the expectation is taken only over $X_{n+1}$, whereas I suppose it also includes averaging over the calibration set.

**Questions:**

* None.

**Limitations:**

* None.

---

> ### Author Rebuttal · Authors · 2024-08-07
>
> Thank you for your thoughtful review and positive evaluation of our paper. We appreciate your recognition of the strengths of our work, particularly the formulation of conformal prediction and the insights gained from our motivating example, as well as the thoroughness of our experiments across different dimensions.
>
> We acknowledge your feedback regarding the readability of the figures and the notation used for expectations, particularly in Equation 2. We agree that clarity in these areas is crucial, and we will make the necessary improvements in the camera-ready version to enhance the paper's overall presentation and precision.

---

> > ### Comment · Reviewer_qXD6 · 2024-08-12
> >
> > I have reviewed the authors' responses and the discussions from other reviewers, and I am confident in maintaining my positive score.

---

### Official Review · Reviewer_vXbJ · 2024-07-13

**Soundness:** 3
**Presentation:** 3
**Contribution:** 3
**Rating:** 5
**Confidence:** 3

**Summary:**

The paper introduces a novel length optimization technique for conformal prediction designed to produce the shortest valid prediction intervals.

**Strengths:**

- The paper introduces a novel framework, Conformal Prediction with Length-Optimization (CPL), which effectively balances the need for conditional validity and length efficiency in conformal prediction.
- Extensive empirical work is provided demonstrating CPL's performance compared to different methods across various real-world and synthetic datasets.
- The paper is overall well written. Beside the theoretical framework, it also provides practical insights into its implementation, which makes it easier to follow.

**Weaknesses:**

- The computational cost of the proposed algorithm seems expensive.
- The length optimization framework dependent heavily on the structure of covariates, which might not be feasible in all practical situations.

**Questions:**

- The equation (2) appears to be equivalent to conditional coverage instead of marginal coverage in Gibbs et al.(2023) [1].
- In figure 2(b), the "optimal" solution is with respect to the choice of score function.
- In chapter 5.3, do you compare with weighted version of split conformal or non weighted?


[1] Gibbs, Isaac, John J. Cherian, and Emmanuel J. Candès. "Conformal prediction with conditional guarantees." arXiv preprint arXiv:2305.12616 (2023).

**Limitations:**

See above.

---

> ### Author Rebuttal · Authors · 2024-08-07
>
> Thank you for your review and for recognizing the strengths of our paper. We appreciate your acknowledgment of the novel framework introduced in the paper,  i.e., Conformal Prediction with Length Optimization (CPL), and the extensive empirical work we conducted to demonstrate its effectiveness. Your feedback on the clarity and practical insights of our writing is also greatly appreciated.
>
> Below we respond to your comments.
>
> **Computational Cost**
>
> Solving minimax problems is a common approach within the machine learning community, particularly in areas such as adversarial learning. In the specific case of CPL, the outer minimization step is computationally lightweight and can be viewed as a form of hyperparameter tuning. For instance, in scenarios focused on marginal coverage, the outer minimization reduces to a simple scalar optimization. Additionally, the outer minimization is linear in terms of its variables, which simplifies gradient calculations. The complexity of the inner maximization is comparable to training a standard machine learning model; e.g. by using standard gradient steps on the parameters. In general, the complexity of the inner maximization is similar to the complexity of training for example a neural network; and CPL requires very simple neural networks--e.g. two layer MLPs, or fine-tuning a pre-trained model. We will ensure to include a comprehensive discussion on scalability in the camera-ready version of the paper.
>
> **Structure of Covariates**
>
> Our framework is designed to work for any given (black-box) predictive model and score function. Doing length optimization in this scenario requires learning features of the data to properly adapt the threshold function for the scores. These features are often present in real world datasets and can be learnt from data as we showcased in our experiments.
>
> **Additional Experiment on LLM Question Answering**
>
> To further demonstrate the applicability/scalability of our method, we will include a large-scale experiment involving LLM question-answering in the camera-ready version (please see the uploaded 1-page PDF file). This experiment uses multiple-choice question-answering datasets, including TruthfulQA, MMLU, OpenBookQA, PIQA, and BigBench. The calibration data for MMLU alone consists of approximately 100,000 prompts and answers, illustrating the large scale of this experiment.
>
> The goal is to quantify the uncertainty of Llama 2 and create prediction sets using this model. We follow a procedure similar to that proposed by prior work [4], adapting it as follows: for each dataset, the task is a multiple-choice question-answering. We pass the question to Llama 2 using a consistent prompt: "This is a 4-choice question that you should answer: {question} The correct answer to this question is:" We examine the logits of the first output token for options A, B, C, and D. Applying a softmax function gives us a probability vector, and we define the conformity score as $1 − \text{probability of the correct option}$
> similar to $1-f(x)_y$ for classification.
>
> CPL is implemented using a linear head (as $h(x)$) on top of a pre-trained GPT-2. In more detail, with GPT-2 having 768-dimensional hidden layers, the inner maximization involves optimizing a 768-dimensional linear map from GPT-2’s last hidden layer representations to a scalar. We also implemented the baseline method that directly applies the split conformal method to the scores. Please **see the uploaded PDF for plots** and a caption for more details.
>
> **Other Comments**
>
> "The equation (2) appears to be equivalent to conditional coverage ..."
>
> It is indeed equivalent to the notion of conditional coverage of Gibbs et al. [1]. As we also cited in our submission, the notion of conditional coverage was first introduced by Gibbs et al. [1]. We built upon their notion and introduced length optimization on top of it.
>
> "In figure 2(b), the "optimal" solution is with respect to the choice of score function."
>
> That is true. We thank the reviewer for bringing up this point. we will clarify this in the revised manuscript.
>
> "In chapter 5.3, do you compare with weighted version of split conformal or non weighted?"
>
> We compare with both split conformal and the method of Gibbs et al. [1]. One can interpret the method of Gibbs et al. [1] as a generalization of weighted conformal prediction in the sense that weighted conformal prediction provides conditional validity with respect to a single covariate shift whereas the method of Gibbs et al. [1] provides conditional validity with respect to a class of covariate shifts.
>
> **References**
>
> [1] Gibbs, Isaac, John J. Cherian, and Emmanuel J. Candès. "Conformal prediction with conditional guarantees." arXiv preprint arXiv:2305.12616 (2023).

---

> > ### Comment · Reviewer_vXbJ · 2024-08-12
> >
> > Thank you for the rebuttal. I have read through it and will maintain my score.

---

### Author Rebuttal · Authors · 2024-08-07

We thank all the reviewers for taking the time to review our submission and providing helpful feedback. We address the reviewers individually below. In addition, a pdf is attached including (i) plots for an additional experiment which we will add to the revised manuscript, and (ii) a sensitivity analysis asked by the reviewer 5vGW.

---

### Comment · Area_Chair_h41Z · 2024-08-12
**Please interact with authors**

Dear reviewers to "Length Optimization in Conformal Prediction" (NeurIPS),

Thanks a lot for your work so far. This is a gentle reminder that you should interact with the authors at this stage. At a minimum, this would involve acknowledging the rebuttal (if you have not done so yet).

Best,
The AC

---

### Decision · Program_Chairs · 2024-09-25

**Decision:**

Accept (poster)

**Comment:**

The present paper considers a framework for length optimization in conformal prediction. The idea is to design a predictive interval for the response variable with the prescribed coverage $1-\alpha$ and with nearly optimal length (this description is somewhat less general than the paper, where responses need not be real-valued). The idea is this: start with a nonconformity score $S(x,y)$ (as is standard) but try to choose the "quantiles" of $S(x,y)$ in an adapted fashion. This requires the introduction of an adequate framework at the population level (allowing for covariance shifts, groups and other things) that is then adapted to finite samples and well-behaved hypotheses classes. That framework is strong related to recent work by Gibbs et al. (arXiv:2305.12616), but the goals are quite different, as that paper does not address length optimization directly.

The reviewers raised many questions during the discussion period about computational efficiency and many other details. Three of them had final scores in the "borderline accept to accept" range (5,6,7) and seemed satisfied with the discussion. The fourth reviewer (xcAK) gave a 4 score (borderline reject): they also seemed satisfied that the discussion helped clarify most points of interest, but felt that the revision required to address these points would be quite substantial. My own evaluation is somewhat at variance with xcAK's: although there are many points to be addressed, I believe that the paper will not be greatly changed. Therefore, I recommend acceptance.